# Relative effects of land conversion and land-use intensity on terrestrial vertebrate diversity

Philipp Semenchuk [1✉], Christoph Plutzar [1,2], Thomas Kastner [3], Sarah Matej [2], Giorgio Bidoglio [3], Karl-Heinz Erb [2], Franz Essl [1], Helmut Haberl [2], Johannes Wessely [1], Fridolin Krausmann [2] & Stefan Dullinger [1]

Land-use has transformed ecosystems over three quarters of the terrestrial surface, with massive repercussions on biodiversity. Land-use intensity is known to contribute to the effects of land-use on biodiversity, but the magnitude of this contribution remains uncertain. Here, we use a modified countryside species-area model to compute a global account of the impending biodiversity loss caused by current land-use patterns, explicitly addressing the role of land-use intensity based on two sets of intensity indicators. We find that land-use entails the loss of ~15% of terrestrial vertebrate species from the average 5 × 5 arcmin-landscape outside remaining wilderness areas and ~14% of their average native area-of-habitat, with a risk of global extinction for 556 individual species. Given the large fraction of global land currently used under low land-use intensity, we find its contribution to biodiversity loss to be substantial (~25%). While both sets of intensity indicators yield similar global average results, we find regional differences between them and discuss data gaps. Our results support calls for improved sustainable intensification strategies and demand-side actions to reduce trade-offs between food security and biodiversity conservation.

[1] Department of Botany and Biodiversity Research, University of Vienna, Rennweg 14, 1030 Vienna, Austria. [2] Department of Economics and Social Sciences, Institute of Social Ecology, University of Natural Resources and Life Sciences, Vienna (BOKU), Schottenfeldgasse 29, 1070 Vienna, Austria. [3] Senckenberg Biodiversity and Climate Research Centre, Senckenberganlage 25, Frankfurt am Main 60325, Germany. ✉email: philipp.semenchuk@univie.ac.at

Land-use (LU) is considered the most important driver of biodiversity loss in terrestrial environments[1], mainly because it shrinks, fragments and degrades natural ecosystems[2]. The resulting human-dominated landscapes often represent mosaics of ecosystems that are used at varied levels of intensity, mixed with remnants of natural ones. The ensuing decline of biodiversity depends on the capacity of species adapted to pristine ecosystems to survive in those reshaped by human use[3,4]. Although it is widely recognized that this capacity depends on both LU-type and LU-intensity[5], the relative impact of LU-intensity on biodiversity is understudied[6–9] and only covered rudimentarily in most global-scale studies[5,10]. However, an understanding of the effects of LU-intensity on biodiversity is critical, as LU intensification is expected to become pivotal in the future due to the increasing demands for LU products and the simultaneous mandate to safeguard remaining pristine ecosystems[10,11]. Due to the multidimensional nature of LU-intensity[9], as well as large data uncertainties related to it[12,13], its impacts on global biodiversity could so far not be quantified satisfactorily. Here, we fill this gap and disentangle the contribution of LU-intensity from total biodiversity losses caused by LU practices worldwide with the help of new methodology.

The countryside species-area relationship (cSAR) is one of the main approaches used for quantifying effects of LU on biodiversity[14,15]. It is specifically tailored to quantify the loss of species with varied pre-adaptations to human-used ecosystems[3,16]. On a global scale, the approach has so far only been applied at the resolution of large regions[17–19], and LU-intensity has only been represented with coarse proxies[9,10]. Moreover, previous applications of the cSAR approach only predicted how many, but not which species face regional extinction. This aspect, however, is crucial for determining how landscape- or regional scale threats to species translate into global extinction risks, a major criterion for rating species on international red lists[20]. Here, we expand the cSAR approach to fill these gaps by increasing the spatial resolution and explicitly including LU-intensity. Based on this approach, we answer how high the landscape-scale contribution of LU-intensity to total biodiversity loss is, and which individual species face regional extinction due to LU.

First, we applied the model to terrestrial vertebrate data on a fine spatial resolution of 5 × 5 arc minutes, based on LU data featuring 45 LU types (aggregated to 6 broad LU types for issues of presentation (SI Table 2)). Second, we developed an approach to include spatially explicit, continuous LU-intensity descriptors at the same resolution in order to separate biodiversity effects of land conversion (i.e. the replacement of pristine ecosystems by various LU types) from those of LU-intensity[15] (i.e. the effects of management without changing the LU type) in both converted and unconverted ecosystems (such as natural grazing land and forests subject to LU). This approach is based on linking model coefficients from ref. [5], who analyse local assemblage data from the PREDICTS database[21], with spatially explicit LU-intensity indicators around the base year 2010 (Methods). We developed two intensity indicator sets based on different input data. Intensity indicator Set 1 represents the human appropriation of net primary production (HANPP), a comprehensive, systemic metric of LU-intensity[12,22] which is based on measuring the effect of LU on the availability of annual biomass flows (trophic energy) in the ecosystem and is calculated for all LU-types[23]. Intensity indicator Set 2 combines input and output metrics, such as fertilizer application rates and livestock densities[12,22] (Methods). We use average effects of LU-intensity across both sets to estimate their general influence on biodiversity, and additionally explore global and regional differences between the two sets. We used the modified cSAR-model not only to compare current land conversion and LU-intensity effects on biodiversity, but additionally quantified, by means of an explorative, counterfactual scenario, the scale of biodiversity effects from further intensification of currently used land in the future. Finally, we used the modified cSAR-model in combination with a randomization approach to assess the effects of LU on global native area-of-habitat (AOH) for each of 5200 mammal, 10,498 bird, 5522 amphibian, and 5730 reptile species (Methods).

Here, we show that LU-intensity has a significant effect on both biodiversity metrics, that further LU intensification without land conversion has a strong potential to increase biodiversity loss, and that different LU-intensity indices show different spatial patterns globally. We conclude that sustainable, biodiversity-friendly intensification methods are advisable if trade-offs between food security and biodiversity conservation are to be reduced, and call for further research to improve our understanding of regional LU-intensity effects.

## Results

**Landscape-scale species loss.** Depending on the set of LU-intensity metrics, current LU patterns entail a global average species loss of 14.6–15.1% (according to intensity indicator Set 1 and Set 2, respectively) from terrestrial landscapes containing areas under LU (i.e., without wilderness landscapes), or 11.2–11.6% including wilderness (see Supplementary Data 1 for more summary statistics and details on different models incl. wilderness). There is, however, substantial spatial variation behind the overall impact of land use on biodiversity (Fig. 1). For instance, calculated loss is higher than 50% on 8.4–8.9% of the global land area. Average loss is somewhat lower for amphibians (11.8–12.5%) than for birds, reptiles and mammals (14.8–15.3%, 14.9–15.5% and 14.4–14.8%, respectively; Supplementary Data 1), but geographical patterns are similar across taxonomic groups (Supplementary Fig. 1). Across LU-types, cropland accounts for 5.6–5.8%-points and pastures for 4–4.3%-points of the calculated species loss, the four remaining LU-types accounting for less than 1.6%-points each (i.e., builtup, grazing land, forests, plantations). Geographical patterns differ widely across LU-types, with, for instance, species loss from cropland being especially pronounced in eastern North America, northern India and parts of Southeast Asia (Supplementary Fig. 2). In almost all grassland and shrubland biomes, LU leads to average species losses lower than the global average (i.e., <14.6%), while species losses in most forest biomes are higher than average (see Supplementary Data 6 for details). Especially tropical dry broad-leafed forests, temperate broad-leafed forests and Mediterranean forests show high species losses, each having, on average, twice as high losses than the global average.

**Effects of LU-intensity.** Of the average species loss of 14.6–15.1%, about one quarter (3.4–3.8 %-points) is caused by LU-intensity while the remaining three quarters (c. 11.2–11.3 %-points) result from conversion of primary ecosystems to various LU-types, especially cropland and pastures (Fig. 2, Supplementary Fig. 2, Supplementary Data 1). Management of unconverted ecosystems contributes importantly to the LU-intensity effect, with livestock grazing on natural grazing land and wood extraction from primary forests contributing 1.54–1.57 and 0.89–0.98%-points to the total LU-intensity effect of ~3.6%, respectively. Although the majority of these unconverted, but used ecosystems are managed with low intensity (Supplementary Fig. 3), their contribution to the total intensity effect is still disproportionately large because of the large area they cover globally (Supplementary Fig. 4). On areas where habitat conversions occurred, LU-intensity effects are lower and contribute 0.9–1.3 %-points to the total loss. The contribution

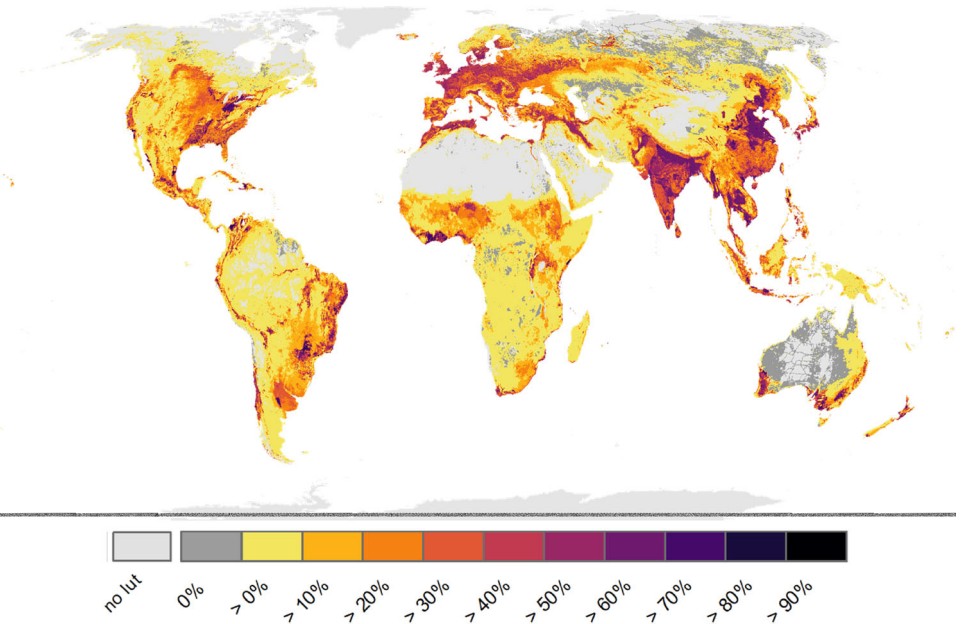

**Fig. 1 Impending loss of terrestrial vertebrate species richness in response to current land-use activities.** Species losses are calculated as sums across all 45 land-use types considered here in 5 × 5 arcmin landscapes and due to land conversion and current land-use intensity together. Numbers are lost percentages of the pristine species richness of each cell, estimated from species range maps and the area-of-habitat approach. Numbers are means of two models with different land-use-intensity indicators (Methods), see Fig. 2 for the effect of land-use intensity alone and the comparison of the two indicator sets. No lut = no land-use type found within the respective landscape.

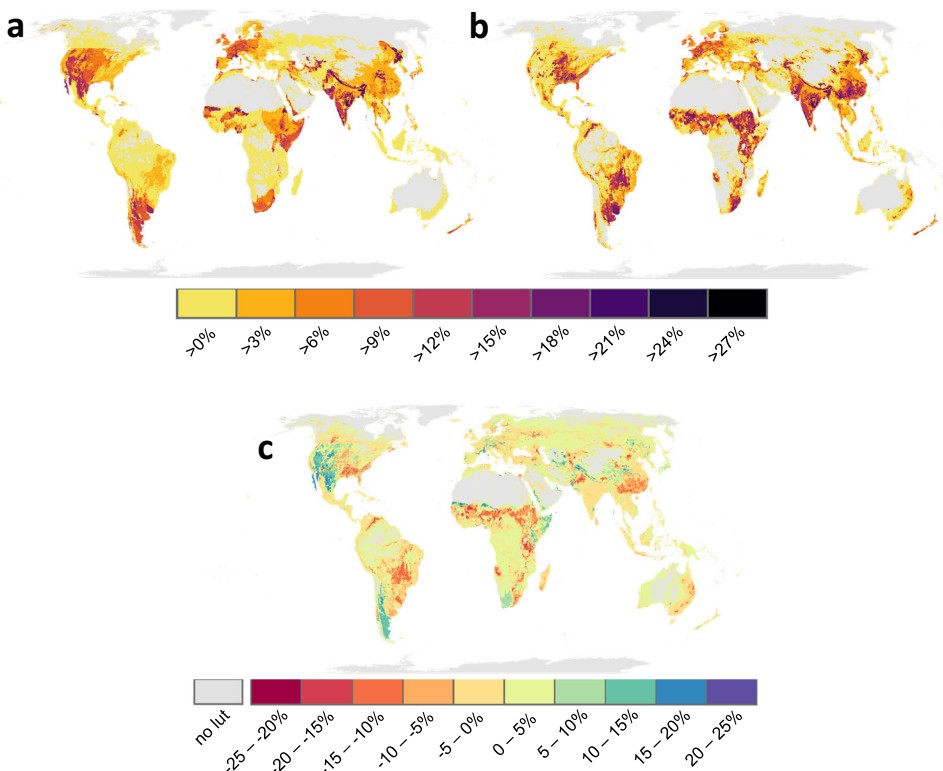

**Fig. 2 Impending loss of terrestrial vertebrate species richness in response to land-use intensity alone.** Calculations are based on (**a**) Set 1 (HANPP) and (**b**) Set 2 (various published sources), while (**c**) shows the differences between (**a**) and (**b**), computed as Set 1 minus Set 2. Species losses are calculated as sums across all 45 land-use types considered here in 5x5 arcmin landscapes due to land-use intensity alone, i.e., without the effect of land conversion. Numbers are lost percentages of the pristine species richness of each cell, estimated from species range maps and the area-of-habitat approach. No lut = no land-use type found within the respective landscape.

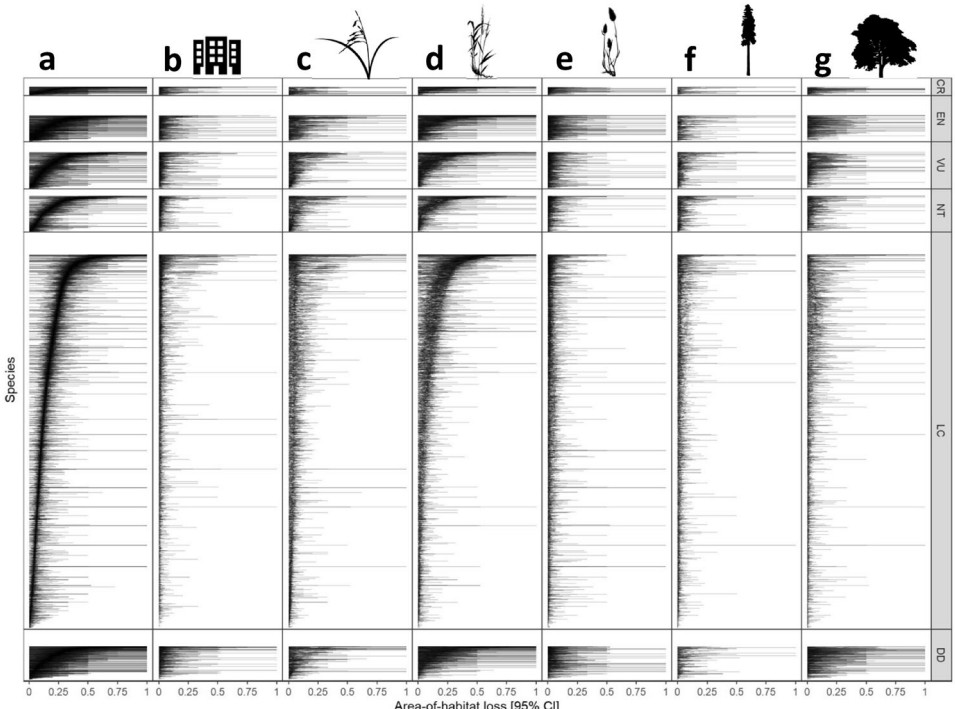

**Fig. 3 Impending global native area-of-habitat reduction of terrestrial vertebrate species in response to current land use patterns. a** All land-use types, **b** Built-up, **c** Cropland, **d** Pastures, **e** Grazing land, **f** Plantations, **g** Forests. Each horizontal line represents the 95 percent confidence limits of the native area-of-habitat loss computed for one species (see Supplementary Data 7–10 for detailed information on each species and Supplementary Information 6 for corresponding figures separated by taxonomic group and conversion and intensity). Within each level of the IUCN Red List classification (vertical groups), species are sorted by their mean area-of-habitat loss from all land-use types (panel **a**). CR Critically endangered, EN Endangered, VU Vulnerable, NT Near threatened, LC Least concern, DD Data deficient (IUCN).

of LU-intensity to calculated species loss is higher in grassland and shrubland than forest ecosystems (Supplementary Data 6).

To explore the scale of potential species losses triggered by future LU intensification, we re-run the cSAR-model under an explorative counterfactual scenario in which we assume that all currently used land is used with high intensity (Methods) while the extent and type of LU remain as they are today. Under these assumptions, the average species loss increases from 14.6–15.1% to 26.6% (Supplementary Data 1), and the impact caused by LU-intensity rises from 3.4 to 3.8 % under current LU-intensity to 15.4% (Supplementary Data 1). Under such a scenario, the largest impact on biodiversity would occur on unconverted forest and grazing land ecosystems under use (Supplementary Data 1).

**Differences between intensity indicators**. The results achieved when running the cSAR model with LU-indicator Set 1 or 2 are highly correlated with each other (linear regression, $R^2 = 0.93$, $F(1, 4182191) = 5.586*10^7$, $p < 10^{-16}$). Nevertheless, differences exist: when considering global averages, calculations with Set 2 yield slightly higher total average species loss caused by higher intensity effects in five of the six LU-types (Supplementary Data 1). These differences are especially pronounced for pastures and cropland, where the intensity effect of Set 2 is about 1.5 and 1.3 times higher than in Set 1, respectively. Besides some similarities between global averages, in individual landscapes species losses can differ up to 25%-points from each other (Fig. 2c). Geographically, calculations based on Set 2 result in higher species loss in eastern China, sub-Sahara Africa, central South America and eastern USA, while Set 1 results in higher loss in northern India and western North America (Fig. 2c).

A detailed exploration of causal factors creating these differences is beyond the scope of this study. Differences may, for instance, result from conceptual uncertainties (e.g. weighting of LU-intensity in the indicator sets), data uncertainties (e.g. inaccuracies of LU- and input/output maps), or simply lack of knowledge or availability of coherent data products. The regional differences of LU-intensity effects we present here suggest that future research on this topic is warranted.

**Consequences for species' native area-of-habitat (AOH)**. Our calculations suggest that species loss due to current LU patterns decreases the global native area-of-habitat (AOH) by 14.1%, on average across all species considered here (Fig. 3). Area-of-habitat losses are significantly lower among species classified as least concern (13.4%, CI 12.8–13.9%) than most other IUCN classes, and highest among those classified as critically endangered (17.1%, CI 13.7–21.1%) (Supplementary Data 3). A decomposition into effects of LU-intensity and land conversion reveals similar contributions as those reported for species richness above, i.e., one quarter (3.2 %-points) and three quarters (10.9%-points), respectively (Supplementary Fig. 6).

At the level of individual species, confidence limits of predicted AOH losses include 100%, corresponding to impending global extinctions, for 556 species (75 mammals, 98 birds, 174 reptiles, 209 amphibians), and 50% for 1673 species (see Supplementary Data 7–10 for tables showing results for each species). However, most of the species with highest AOH losses have small native AOH, leading to wide confidence intervals of the randomization method used for that group (Fig. 4). Interestingly, predicted AOH loss peaks for geographically relatively restricted species, but levels out or decreases again for species with smallest AOH (Fig. 4).

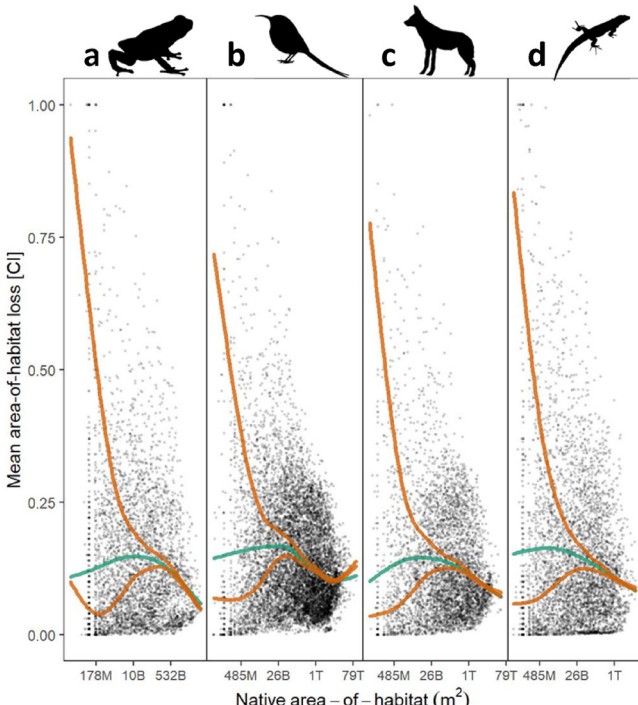

**Fig. 4 Calculated area-of-habitat loss against native area-of-habitat of each species per taxonomic group. a** Amphibians, **b** Birds, **c** Mammals, **d** Reptiles. Green line = spline on mean loss (black dots), orange lines = splines on lower and upper CI of loss (cf. Fig. 3). Please note the log-scale on the x-axis denoting the species' native area-of-habitat in m².

Among LU-types, differences are statistically significant and pastures trigger the strongest AOH reductions (5.9%-points, CI 5.6–6.2%-points), followed by cropland (3.1%-points, CI 2.8–3.3%-points) (Supplementary Data 3).

## Discussion

Our results indicate that current land use causes the loss of ca. 15% of vertebrate species richness from the average used terrestrial landscape across Earth, with peaks of more than 50% on 8.4–8.9% of the terrestrial land surface. We further show that LU-intensity contributes about one-quarter to this loss in addition to the losses caused by land conversion. Irrespective of the data uncertainties discussed here, our findings highlight a substantial contribution of LU-intensity, in particular in light of the large fraction of land which is still used with low intensity (Supplementary Data 3). As a corollary, the scale of further biodiversity impacts from future intensification can be considerable: the presented counterfactual intensification scenario yields a four-fold increase of current species loss caused by LU-intensity alone. While this scenario is hypothetical and explorative by nature and not meant to reproduce realistic trajectories, it is useful to identify upper boundaries and the potential present in future LU intensification.

The species loss metrics calculated here refer to an eventual equilibrium situation between the habitat mosaic created by LU and the number of native species able to persist in this mosaic, i.e., to a situation in which all extinction debt has been paid off[24]. However, remnant populations may persist for a long time, and individuals from neighbouring source populations may help maintaining vital populations in an otherwise unsuitable landscape. Also, our results refer to the impending loss of species adapted to the pristine ecosystems of these landscapes only[25], while they do not account for possible immigration of novel, non-native species adapted to the landscapes created by LU[26] from

outside a focal region or landscape. In summary, net realized species loss, as measured through e.g., field surveys, may be lower than the predictions of our model. From the perspective of biological conservation, the focus on native species loss is particularly relevant, because species turnover resulting from LU often induces the replacement of specialized and/ or geographically restricted species by widespread generalists leading to biological homogenization[27]. However, disregarding immigrating species may contribute to an overestimation of species richness change by the cSAR model. On the other hand, the focus on pristine species pools and their depletion excludes the fate of species that had immigrated into a focal landscape in response to historical LU centuries or even millennia ago. These species are often considered native today even if they were not present prior to the historical introduction of LU in a given landscape. In areas with a long LU history[28,29] these species may actually represent a major part of those extirpated by recent LU-intensification. In such areas, for instance in European cultural landscapes, the potential impact of LU intensification on species loss is hence certainly even greater than the numbers calculated here indicate.

In contrast to the cSAR model, species lists from local assemblages do include, to an unknown extent, species present because of unpaid extinction debt or immigration in response to human usage. Despite these differences, the global average of species loss calculated here is similar in magnitude to the one reported by a recent meta-analysis of such local assemblage data[5]. This similarity may in part be due to the fact that we used model coefficients from this study to quantify the effect of LU-intensity in our cSAR model (Methods). Nevertheless, the bulk of our calculations was based on data sources other than ref. [5] and the results are, hence, independent. Further, LU-intensity only contributes a quarter to our results while the remainder is driven by land conversion which has been parameterized in a completely different way. From the agreement of the two studies we tentatively conclude that a value of ca. 15% can be considered a plausible, robust estimate of the average magnitude of LU-driven species loss in current terrestrial environments.

Land-use intensity is a complex, multifaceted phenomenon with numerous possible direct and indirect effects on biodiversity. As a consequence, it is difficult to represent it by one comprehensive surrogate indicator in biodiversity models[9]. Here, we account for this conceptual problem by using two different LU-intensity indicator sets (Fig. 2), one representing a comprehensive, systemic metric (HANPP, Set 1) and the other one combining a number of input-output metrics (Set 2). While Set 1 uses an intensity metric consistent across all LU-types, Set 2 combines different metrics compiled from independent datasets focused on individual LU-types. Although calculated effects on biodiversity are largely congruent with these two sets, especially with respect to the global average, regional differences underpin important uncertainties corroborating calls for careful consideration of how LU-intensity is operationalized in biodiversity research[9,12]. These uncertainties include general lack of knowledge about how and by which processes LU-intensity affects biodiversity, but also issues of data quality[30] or of the assumptions necessary when combining different data products (see Methods for details). Against the background of these uncertainties and our results, combining or comparing different sets of LU-intensity indicators appear generally advisable for analysing biodiversity impacts of LU. Such a comparison makes existing uncertainties explicit and represents the range of possible effects according to available data and the state of knowledge on biodiversity-LU-intensity relationships.

We show that the risk of global extinction caused by LU is highest for geographically restricted species. Both the mean calculated AOH losses, as well as their upper confidence limits (and the confidence intervals' widths) are higher for species with smaller

native AOH (Fig. 4). A recent study found that grid-cell level persistence probabilities are lower for small-ranged species[31], i.e., that those species are more likely to disappear from a landscape in response to LU change than wide-ranged ones. This puts species with small native AOH under double jeopardy, as they (a) have higher landscape level extinction probabilities and are simultaneously (b) more likely to lose their entire AOH and go globally extinct. However, in our simulations, predicted AOH loss peaks for species with small native AOH, but levels out or decreases again for species with smallest AOH (Fig. 4). This may indicate that the AOH of these species are situated in protected areas where little to no LU occurs.

The development of future LU patterns and their effects on biodiversity is of particular interest for biodiversity conservation planning[4]. Concomitant with mandates to stop the loss of biodiversity[32] and increase restoration efforts[33], achieving future food security for a growing human population and providing sufficient biomass for a growing economy will in large part rely on intensification of ecosystems already under use[2], as little fertile land is still left unused[34,35]. Our results suggest that LU-intensity at current levels already contributes about a quarter to both native species richness loss and AOH reductions. However, our results also suggest that further intensification has the potential to almost double average impending species losses, even if future biomass demand is to be sustained without further conversion of native ecosystems to LU. Avoiding further land conversion certainly remains of prime importance, especially in regions with no LU history, or a past LU that has not resulted in documented biodiversity loss over the last centuries, such as in remaining primary rainforests[36]. In addition, however, detrimental effects of intensification, and especially of 'conventional intensification', are well documented[5,37], especially in regions with a longer history of low-to-medium LU-intensity[29,38]. Strategies that focus on halting land conversion hence appear insufficient for safeguarding biodiversity, and we highlight the importance of further developing sustainable intensification strategies which simultaneously spare pristine ecosystems from conversion and protect or even increase biodiversity in human-used ecosystems[39].

In addition, our results support recent calls for demand-side action to limit primary biomass demand. Such strategies include changes of consumption patterns towards more plant-based diets, reductions of losses in supply chains, or reduction of consumption of non-food biomass such as furniture or fuel. The strong LU-intensity impact also warrant serious cautions against climate change mitigation strategies such as bioenergy with carbon capture and storage (BECCS), as envisaged in many IPCC scenarios, because they might require extensive land resources and yield considerable pressure for further LU intensification.

Scenarios of future LU-intensity and their impact on biodiversity at sufficient detail are largely lacking to date[40]. However, given the considerable current and likely increasing future importance of LU-intensity, the development of such scenarios appears pivotal to provide relevant information for decision makers integrating goals along the nexus of food security, climate protection and biodiversity conservation. The present study represents a first step in that direction, and the methods presented here may be vital tools to asses biodiversity impacts of such detailed LU scenarios, once they become available.

## Methods

**cSAR model.** We used the numerical cSAR model[16] to calculate native species loss of four taxonomic groups (mammals, amphibians, reptiles, birds) caused by 45 LU types that were mapped onto a reference 5 × 5 arcmin grid (we also call individual grid cells landscapes in the following) of the global land area excluding Greenland and Antarctica. Calculations were based on (a) gridded LU-intensity and LU-type information (see below), (b) effects of LU-intensity on species richness derived from recently published meta-analyses[5,21], and (c) information on species

distributions and habitat affiliations from IUCN and Birdlife International databases[41,42]. For presentation of results, we aggregated the calculated effects of the 45 LU-types into those of six broad LU-types (cropland (30 annual crop types); pastures (non-grassland converted to grassland); grazing land (natural/ near-natural areas with livestock grazing); builtup (sealed areas); plantations (11 permanent crop types plus timber plantations), and forests (natural/ near-natural forest under forestry); see Supplementary Data 2 for details).

In the below formulae, we use the following indices: $g$ = taxonomic group, $n$ = grid cell, $b$ = broad LU-type. We calculated the total number of native species losses for each taxonomic group g and grid cell n as

$$S_{g,n}^{loss} = S_{g,n}^{pot} \times \left( 1 - \left( \frac{A_n^{cur} + \sum_{b=1}^{b_n} h_{g,n,b} \times A_{n,b}}{A_n^{pot}} \right)^{z_n} \right) \quad (1)$$

Here, $S_{g,n}^{pot}$ is the potential species richness in pristine ecosystems, $A_n^{cur}$ is the pristine ecosystem area where no LU occurs (in m²), $A_n^{pot}$ is the grid cell's terrestrial area (in m²), $h_{g,n,b}$ is the affinity parameter, $A_{n,b}$ is the area of the LU-type, and $z_n$ is the grid cell's SAR exponent taken from ref. [43]. The model's components are described below.

**Potential species richness** $S_{g,n}^{pot}$. We defined potential species richness of a landscape as the number of species for which the area-of-habitat (AOH) under pre-human or pristine conditions overlap the landscape (here referred to as native species and native AOH). Following ref. [44], we used range maps of all mammal, reptile and amphibia species provided by the IUCN[45] and bird species by Birdlife International[46] databases to calculate gridded species richness via, first, overlapping each species' range polygons with a 5 × 5 arcmin reference raster, second, constraining the resulting list of species per raster cell to those adapted to the pristine ecosystem(s) of these raster cell as defined in ref. [25], and, third, constraining the resulting list of species by each species' elevational range, also provided by the IUCN[42]. Here, we are interested in the total historical range of extant species[46] and hence included all parts of the range where the species were indicated as (i) Extant, Probably Extant, Possibly Extinct, Extinct and Presence Uncertain, (ii) Native and Reintroduced, and (iii) Resident or present during the Breeding Season or the Non-breeding Season, in the cited data sources.

We first rasterized each species' range polygons using the raster and fasterize packages in R[47]. Second, for each terrestrial grid cell in our reference raster, we created a species list by extracting each species' gridded range using the velox package in R. Third, we ascertained that each cell's species list contained only terrestrial species by excluding species which exclusively have aquatic habitat affiliations. The species' habitat affiliations were directly taken from the IUCN and Birdlife databases[42,46]. Fourth, we removed species from this cell's species list which, according to the IUCN, are not affiliated with that cell's pristine ecosystem. We therefore manually assigned the habitats distinguished in the ICUN habitat affiliation scheme to one or several of the 14 broad ecosystem types distinguished and mapped in ref. [25] (Supplementary Data 4). The maps in the referenced study "approximate the original extent of natural communities prior to major land-use change"[48] and, hence, represent pristine ecosystems or potential vegetation types. Fifth, we excluded species whose elevational range did not overlap the elevational range of the grid cell using the GMTED2010 dataset (www.usgs.gov). These refinement steps were taken because species' range maps usually deliver coarse-scale extent of occurrence rather than AOH information[44]. Finally, we counted the species identities in each grid cell as $S_{g,n}^{pot}$. The species lists created in this step were also used for later steps, referred to in the appropriate sections.

**Areas of pristine ecosystems** ($A_n^{cur}, A_n^{pot}$) **and LU-types** ($A_{n,b}$). The potential pristine ecosystem area $A_n^{pot}$ is defined as the cell's entire terrestrial area (excluding water bodies as defined by the land mask of the HYDE 3.2.1 database[49]). As the area of pristine ecosystems currently found in each grid cell ($A_n^{cur}$), we used the proportion of $A_n^{pot}$ marked as wilderness and non-productive/ snow areas as described below. The area of each of the 45 LU-types within each grid cell ($A_{n,b}$) was extracted from respective land cover and LU maps applying the approach outlined in ref. [50] – with 2010 as year of reference wherever possible (Supplementary Data 2).

Following ref. [50], builtup land, total cropland (including annual and permanent crops/ plantations), permanent pastures (areas used as pastures for more than five years) and rangeland (available in the two sub-categories natural and converted) extents were taken from the LU database HYDE 3.2.1[49] which was adapted to include rural infrastructure areas by assigning 5% of each grid cell's cropland area to builtup land. We then split the total cropland cover into areas used for 41 different annual and permanent crops by integrating data from the Spatial Production Allocation Model (SPAM) for 2010[51,52] and adjusting them to cropland extent in the data from ref. [50]. To comply with the IUCN habitats classification scheme[42], some of these crops were grouped into the plantation category (permanent crops), while the remainder was grouped into the cropland category (annual crops; see Supplementary Data 2 for details).

Wilderness areas were derived from the combination of human footprint data, i.e., a spatially explicit inventory of human artefact density available for 1993 and 2009[53,54] and intact forest landscape data for 2000 and 2013[55]. Core wilderness areas without human use were defined as having a value of zero human footprint

and, in forests, being part of an intact forest landscape[55]. Within forests, the additional category of peripheral wilderness was introduced for areas where either only zero human footprint is recorded, or only an intact forest landscape exists.

The area remaining in each grid cell after allocating the above land cover types represents area covered by used forests and other land with mixed land uses[56]. Hence, in addition to the approach in ref. [50], forests were split into deciduous and coniferous forests based on the description of the ESA CCI land cover categories[57]. This distinction was necessary for the differentiated allocation of wood harvest (see below). A further refinement was applied by identifying plantation forests, defined as areas in non-forest biomes converted to forests for forestry and areas in forest biomes converted to non-native forest types[58], which were linked to the IUCN habitat class plantations (Supplementary Data 2).

As in ref. [50], the remaining area not allocated to any of the land cover or LU types above is denoted as "other land, maybe grazed"[56]. These lands, typically treeless or bearing scattered tress, were allocated to converted grasslands on areas that potentially carry forests or to natural grassland on areas where the potential vegetation would not consist of forests[25].

To arrive at the six broad LU-type aggregates compatible with the IUCN and Birdlife habitat affiliation schemes[42,46] and PREDICTS categories[21] (needed for quantifying LU-intensity effects, see below in section "Affinity parameter" for details), we rearranged and aggregated the described LU layers as needed (see Supplementary Data 2 for an overview). (a) Builtup remained as described above. (b) Cropland was defined as annual crops, covering the respective 29 SPAM categories plus fodder. (c) Pastures were defined as areas where pristine ecosystems were converted to grasslands and includes permanent pastures and converted rangelands from HYDE 3.2.1[49], plus those parts of "other land maybe grazed" located in forest[25]. (d) Grazing land was defined as natural or near-natural areas where grazing occurs and includes natural rangelands from HYDE 3.2.1[49], plus 50% of each grid cell's open forest area and 25% of each grid cell's peripheral wilderness area, the latter two assumed to be only occasionally grazed and hence given low grazing intensity (see below), plus those parts of "other land maybe grazed" located in non-forest[25]. (e) Forests were defined as forests where forestry occurs and includes 100% of each grid cell's closed forest area, 50% of each grid cell's open forest area, and 25% of each grid cell's peripheral wilderness area, the latter two assumed to be only occasionally used for forestry and given low intensity (see below). (f) Plantations were defined as areas where pristine ecosystems were converted into plantation-like LU and include the 11 SPAM categories representing permanent / plantation crops, plus used forests identified by ref. [58] as plantations (see above). As stated above, these aggregated broad LU-types were needed to align the different LU categorizations used in the different data sources with each other. The effects on biodiversity were then calculated on each of the 45 LU-types and afterwards aggregated to the six broad LU-types to give a better overview.

**Continuous LU-intensity indices ($LUI_{n,b}$).** We constructed continuous LU-intensity indices $LUI_{n,b}$ for each of the 45 LU-types based on gridded management descriptors[15]. For this purpose, we used two different sets of intensity indicators (called Set 1 and Set 2) to compare and combine their impact on predicted species loss. We used two indicator sets to account for the multidimensional nature of LU-intensity[9,12] and to include a wide range of available data products. For an overview of which data products and assumptions went into the individual sets, please refer to Supplementary Data 2.

**Set 1.** Set 1 is taken from the human appropriation of net primary production (HANPP) framework, a socioecological indicator basically describing the LU mediated extraction of biotic resources in the context of global biogeochemical cycles[23]. We used the ratio of $HANPP_{harv}$ to $NPP_{pot}$ as a systemic metric to assess LU-intensity[12,22], with $HANPP_{harv}$ being harvested or extracted biomass and $NPP_{pot}$ being NPP of potential natural vegetation, i.e. the vegetation existing under current climate conditions in the hypothetical absence of LU[23]. The ratio $HANPP_{harv}/NPP_{pot}$ relates harvest to the productivity potential of the land where the harvest takes place and is, thus, robust against geographic differences in natural productivity. As it is related to energy availability in ecosystem food chains, it may be linked to the species-energy relationship, the strongest correlate of spatial biodiversity patterns at larger scales[59].

For calculating $NPP_{pot}$, LPJ-GUESS[60] version 4.0.1 was used in its standard configuration but with nitrogen limitation disabled and forced by the CRU-NCEP climate data[61,62] aggregated from 6-hourly to monthly fields.

$HANPP_{harv}$ of all LU-types except builtup was calculated based on the FAOSTAT database by principally accounting total biomass flows via conversion and expansion factors as outlined in ref. [63]. As a special case, $HANPP_{harv}$ of builtup was assumed to be half of the actual NPP, which was defined as 1/3 of the potential vegetation in ref. [64]. This results in a constant intensity on built-up land of ~17% of $NPP_{pot}$.

$HANPP_{harv}$ of permanent and non-permanent crops was spatially downscaled following 40 permanent and non-permanent crop-specific production patterns from the Spatially-Disaggregated Crop Production Statistics Database (SPAM[52]), merging minor SPAM categories such as "robusta coffee" and "arabica coffee" to ensure consistency with FAOSTAT reporting. Additionally, we added the LU-type fodder, which was downscaled following $NPP_{pot}$ patterns.

Harvest of natural and plantation forest is reported by FAOSTAT in the four categories industrial roundwood, wood fuel, and coniferous and deciduous. We allocated industrial roundwood harvest to closed forests, while we split wood fuel harvest in proportion to productivity between closed and open forests, independently for deciduous and coniferous forests, respectively. For Set 1, we assumed forestry harvest to follow the patterns of forest $NPP_{pot}$[65]. These intensity definitions were used for both natural and plantation forest.

Reported harvest on grazing land and pastures was allocated following patterns of aboveground NPP accessible for grazing as reported in ref. [63]. Following the assumption that systems with low natural productivity allow for a lower maximum harvest than systems with high productivity, we assigned a maximum harvest intensity of 40% at a level of accessible NPP of 20 gC/m² and increased this linearly to a maximum grazing intensity of 80% at 250 gC/m². Such, harvest was concentrated on grazing land and pastures with high productivity. In cases where the calculated national grazing land and pasture harvest demand surpassed NPP availability on grassland, we used information on fertilization rates on grassland[66] to either adjust NPP or harvest data: NPP was boosted in countries where more than 5% of overall fertilizer consumption was applied to grasslands, while countries where no relevant fertilization of grasslands occurred, the reported harvest demand was reduced accordingly, assuming it will be met from other sources. This intensity definition was applied to both (natural) gazing land and (converted) pastures.

**Set 2.** For the LU-intensity indicator Set 2, we used published data from different sources. For cropland we used the input metric nitrogen application rates (in kg N/ha of cropland)[12,22], available for 17 major non-permanent crops[67,68]. For crops from the SPAM categories (see above) not covered by these data, we used the within-grid-cell area-weighted average of other crops in the same cell. For areas designated as cropland in our data (see above) but not in the available N application data, we assumed national average values of the respective crop.

For pastures and grazing land, we used gridded livestock information[69]. We used information on the typical weight per animal to calculate livestock units[70] and aggregated the data for all ruminant species (buffalo, horses, cattle, sheep, goats). This data on livestock numbers per grid cell was then divided by land area per grid cell to arrive at livestock densities, which were applied to the extent of grazing land and pastures. Please note that this dataset contains information on the number of livestock (per species group) per area in a grid cell and thereby differs from the grazing intensity metric applied in Set 1[71], as grazing animals may be fed from other sources than grassland[72].

For builtup, we aggregated a 1 km built-up area density map for 2014[73] to the target resolution of 5 arc min and used it as as intensity indicator.

For natural and plantation forest, we used the same data as described above for Set 1, but we assumed forestry harvest to follow another pattern. We calculated the difference between potential and actual biomass stocks[74] and allocated forestry harvest within each country according to these patterns, i.e., the share of national forest harvest allocated to a forest cell corresponds to its share in the national difference between potential and actual biomass stocks.

**Scaling of LU-intensity indices.** For the purpose of applying linear functions on species richness loss caused by LU-intensity (see below, affinity parameter), we scaled each LU-intensity indicator to values between 0 and 1, with 0 being no intensity (hypothetical) and 1 being the intensity threshold above which an increase of intensity causes no further increase of species loss. This threshold is not necessarily the highest recorded value of an intensity indicator, as effects may be regionally variable. We therefore winsorized some LUI indicators to that intensity threshold before scaling them (dividing by this threshold). These thresholds were defined as follows.

In Set 1, maximum intensity was assumed to be reached at harvesting 100% of $NPP_{pot}$ on cropland.

In forests (natural and plantation), maximum intensity was derived from ref. [75], which limits sustainably harvestable aboveground biomass in forests to 30% of NPPpot. In concordance with the HANPP framework, we included the belowground biomass destroyed by forestry using biome-specific factors[76].

On grazing land and pastures, maximum intensity was defined as removal of all NPP accessible for grazing. This considers only the aboveground and non-woody parts of $NPP_{pot}$. The maximum removable aboveground share was estimated as 50% of $NPP_{pot}$, and the proportion of non-woody vegetation was estimated as 30% (in closed-canopy land cover types) or 100% (on open land cover types)[71]. $HANPP_{harv}/NPP_{pot}$ was assumed to be at its maximum intensity level when the maximum level of grazing intensity, as described above, was reached. The resulting thresholds are in line with literature[77,78], and assume that maximum intensities will be reached faster in systems with low natural productivity.

In Set 2, for all crop types (permanent and non-permanent) except legumes, N application rates were capped at 150 kg N/ha, i.e., we assumed that 150 kg N/ha was the maximum LU-intensity on cropland, beyond which no further species richness loss occurs, i.e., after which an increase of N application rates causes no further increase in species loss based on ref. [79]. For legumes, under the assumption that they need less N fertilizer due to their N-fixing capabilities, we assumed the following cap values, based on information provided in ref. [80]: beans and lentils at 110 kg N/ha, chickpeas at 100 kg N/ha, soybean at 70 kg N/ha and cowpeas, pigeon peas and other pulses at kg N/ha 90.

For pastures and grazing land, maximum intensity was defined as the per biome 80th percentile of livestock-density.

The intensity of builtup area was not winsorized.

**Affinity parameter ($h_{g,n,b}$).** The affinity parameter $h_{g,n,b}$ can be regarded as a LU-intensity dependent weighting factor for the area of each of the 45 LU-types used here. For low affinity, i.e., a small fraction of native species is left due to LU, the area of this LU-type ($A_{n,b}$ in formula 1) is down-weighted, resulting in higher species loss $S_{g,n}^{loss}$ (and vice versa). The affinity parameter consists of two terms, (a) $r_{g,n,b}$, the fraction of species affiliated with a given LU-type, and (b) $f_{n,b}$, the fraction of $r_{g,n,b}$ that remains when LU-intensity ($LUI_{n,b}$) rises to a particular level.

The fraction of species affiliated with a certain LU-type under minimal $LUI_{n,b}$ ($r_{g,n,b}$) is based on the habitat affiliation information taken from the IUCN Red List API[45] and BirdLife data[46] cross-tabulated with our mapped LU-types (Supplementary Data 2). We calculated $r_{g,n,b}$ by dividing the number of species affiliated with a certain LU-type ($S_{g,n,b}^{pot\ LU}$) by the number of native species expected in this cell under pristine ecosystem conditions ($S_{g,n}^{pot}$) as

$$r_{g,n,b} = \frac{S_{g,n,b}^{pot\ LU}}{S_{g,n}^{pot}} \qquad (2)$$

Please note that for the two unconverted broad LU-types grazing land and forests, respectively (see above), we assumed no land conversion prior to its use, leading to $S_{g,n,b}^{pot\ LU} = S_{g,n}^{pot}$. We further assumed that the whole fraction of LU-type affiliated species $r_{g,n,b}$ are present in a given LU-type as long as $LUI_{n,b}$ is minimal (here < 0.17, see below). If $LUI_{n,b}$ is higher, $r_{g,n,b}$ will be further reduced according to functions which vary across LU-types ($f_{n,b}$). These functions were derived from estimates of linear mixed effects models published in a recent analysis of fractional species richness declines in local assemblages. The authors used the PREDICTS database and distinguished four broad LU-types and three categorical intensity levels (Minimal, Light, Intense)[5,21]. The four broad LU-types from this study were assigned to the six broad LU-types we used in our aggregation (see Supplementary Data 2). Note that while cSAR calculations were performed on LU maps that distinguished all 45 LU-types, the intensity-driven loss functions were identical to all types belonging to one of the four categories distinguished in PREDICTS, e.g., for all 30 annual crop types assigned to the cropland category or for all 11 permanent crop types plus timber plantations assigned to the plantation category (see Supplementary Data 2 for details).

Here, we assume that species loss associated with a particular LU-type under Minimal intensity is equal to $r_{g,n,b}$, and that $f_{n,b}$ for the categories Light and Intense is the proportional SR decline relative to Minimal,

$$s_{n,b} = \frac{Est_{int}}{Est_{Minimal}} \qquad (3)$$

with $s_{n,b}$ the fraction of species loss caused by the respective intensification step (from Minimal to Light or Intense, respectively), $Est_{int}$ the published species loss estimates for each of the three categorical intensity levels and $Est_{Minimal}$ the respective published estimate for Minimal (Supplementary Fig. 5). To create a continuous relationship between $LUI_{n,b}$ and $f_{n,b}$, we assigned these three categorical intensity levels to fixed continuous $LUI_{n,b}$ values (with $LUI_{n,b}$ scaled between 0 and 1) and linearly interpolated between them. The definition of the three intensity levels in the literature does not allow an exact assignment along the continuous $LUI_{n,b}$ gradients and we took the most parsimonious assumption that Minimal is between the 1st and 33rd (i.e., at 0.17), Light between the 34th and 66th (i.e., at 0.5), and Intense between the 67th and 99th (i.e., at 0.83) percentile of the scaled intensity gradient. We then placed these $s_{n,b}$ values in the respective coordinate system (see Fig. ED5) and linearly interpolated between the values to yield an intercept and slope for each interpolation step. This procedure allowed us to calculate as

$$f_{n,b} = int_b + slope_b \times LUI_{n,b} \qquad (4)$$

For the extrapolation below $LUI_{n,b} = 0.17$ (i.e., below Minimal), we decided to keep the relative species richness constant because the highest possible $f_{n,b}$ must be 1, in accordance with the IUCN habitat affiliation scheme. In other words, we assumed that all species adapted to a certain LU-type according to the IUCN habitat affiliations can thrive in the respective ecosystems as long as the LU-intensity is minimal. For relative intensity levels >0.83, we argue that extrapolation outside the measured intensity range is uncertain, and that an increase in LUI above 0.83 (i.e., Intense) might not necessarily result in even stronger effects on SR. See Supplementary Fig. 5, which illustrates the results of these considerations and shows the continuous effect of $LUI_{n,b}$ on SR used in this study.

The affinity parameter $h_{g,n,b}$ was then calculated as follows and inserted into formula 1 (cSAR model).

$$h_{g,n,b} = \left(r_{g,n,b}\right)^{1/z_n} \times \left(f_{n,b}\right)^{1/z_n} \qquad (5)$$

**Species loss caused by LU-intensity ($S_{g,n}^{lossint}$).** In order to calculate the relative impact of LU-intensity on species richness, we re-ran the model with $LUI_{n,b} = 0$ in all grid cells and LU types, thereby effectively setting $f_{n,b} = 1$ and $h_{g,n,b} = r_{g,n,b}$. The results of this model can be considered as delivering the land conversion effect without any possible enhancement by intensification. In addition, we designed a hypothetical, back-of-the-envelope intensification scenario where $LUI_{n,b} = 1$ in all grid cells and LU-types.

The contribution of intensity to the species richness loss was then calculated as

$$S_{g,n}^{lossint} = \left(S_{g,n}^{loss} - S_{g,n}^{loss\ conv}\right)/S_{g,n}^{pot} \qquad (6)$$

With $S_{g,n}^{loss\ conv}$ being the results of the $LUI_{n,b} = 0$ model and $S_{g,n}^{loss}$ from Eq. (1).

**Native area-of-habitat loss of individual species.** The cSAR model calculates by how many species the native species pool is reduced in response to LU in each 5 arcmin grid cell. However, it does not identify the individual species lost. To estimate each species' native AOH loss, we randomly drew the predicted number of species lost from the native species pool of each cell.

First, we rounded the number of species lost as calculated by the cSAR model to the next integer for losses from both conversion ($S_{g,n}^{loss\ conv}$) and intensification (here taken as $S_{g,n}^{loss} - S_{g,n}^{loss\ conv}$, see section above: "Species loss caused by LU-intensity"). To avoid rounding all values below 0.5 to 0, and, hence, to underestimate low levels of species loss, particularly in species-poor regions, we used a two-step rounding routine. First, prior to actual rounding, we randomly decided whether a number is rounded to the next higher or lower integer, with the likelihood of either decision depending on the decimal number's (positive or negative) distance to 0.5 (i.e., the decimal number gave the likelihood of rounding up). Second, we took the species list used to generate $S_{g,n}^{pot}$ (see above under potential species richness) and modified it to either contain only species affiliated or unaffiliated with each LU-type, yielding two species lists for each grid cell and LU-type, respectively. The list of species affiliated with a particular LU-type was then used to select species predicted to get lost due to intensification, while the list of species not affiliated with it was used to select species lost due to conversion.

From each grid cell, we then randomly drew as many species from these lists as determined by the rounding routine above, considering each LU-type and whether the number of lost species was caused by intensification or conversion. However, in each cell, each species could only be drawn once, independently of whether it was affiliated with several LU-types. As a consequence, the order in which LU-types are considered when drawing species is relevant for the outcome of the calculation. For instance, species simultaneously unaffiliated with cropland and affiliated with natural forest may never be drawn in response to intensification of natural forest if losses due to conversion into cropland are always handled first. Therefore, we randomly iterated the sequence by which LU-types were considered, i.e., the order of LU-types, in the random draw routine in each of 100 repeated runs.

We repeated the random-draws 100 times to yield a representative sample and processed the resulting 100 lists of species-per-cell losses in the following way. For each of the 100 runs, we summed the areas of all cells each species was drawn from, i.e., predicted to be lost, across all LU-types and within individual LU-types, yielding 100 area sums per species (one per run). From these 100 area sums, we calculated the mean and the 0.025th and 0.975th quantiles as 95% confidence intervals (CIs). The means and CIs were then divided by the species' global AOH (sums of cell areas in native range), thereby yielding the proportional global AOH loss attributable to current LU in general, and to different LU-types or land conversion vs. LU-intensity in particular.

**Description/ presentation of results.** All cSAR model calculations were based on global land use maps that distinguish 45 LU-types as described above. For the sake of simplicity, we present results aggregated to the six broad LU-types cropland, pastures, natural grazing land, built-up, plantations, and forests (natural/ near-natural forest under forestry; see Supplementary Data 2). All calculated SR decreases are expressed in percentage losses relative to $S_{g,n}^{pot}$.

Summary statistics mentioned in the text and Supplementary Data 1, 5 and 6 were calculated as follows. Global, biome-wide and nation-wide average species losses due to conversion, LU-intensity or both were calculated as cell-area weighted means across all cells with native terrestrial vertebrate species either excluding or including wilderness areas (which, for this purpose, are defined as cells where the sum of all LU area equals 0). The percentual land area exceeding a certain threshold of calculated SR decline were calculated by dividing the area sum of all cells exceeding that threshold by the area sum of all cells with native species excluding wilderness.

Differences among average AOH losses (across all taxonomic groups) mentioned in the text and Supplementary Data 3 were modelled using generalized linear models assuming a binomial distribution (proportional AOH loss between 0 and 1), each species' mean AOH loss (mean of 100 random draw runs) as response, and either (a) IUCN categories, (b) land use types, or (c) taxonomic group as predictor variables. Differences between predictor variable levels were then alculated by multiple comparisons via p-values adjusted with the Tukey method. A p-value of < 0.05 was taken as statistically significantly different. We used the function glm from base R[81].

**Reporting summary**. Further information on research design is available in the Nature Research Reporting Summary linked to this article.

## Data availability

The LU-intensity indicator Set 1 data generated in this study have been deposited in the Zenodo database under https://doi.org/10.5281/zenodo.5761990. These data are available under restricted access for reasons of separate future publication, access can be obtained by contacting the corresponding author to discuss the suitability of the data for the project in question and co-authorship of relevant authors. The processed global land-use type distribution and area data processed in this study have been deposited in the Zenodo database under https://doi.org/10.5281/zenodo.5761990. These data are available under restricted access for reasons of separate future publication, access can be obtained by contacting the corresponding author to discuss the suitability of the data for the project in question, co-authorship of relevant authors and details about the various datasets used to create these data. The raw data generated in this study on species richness loss as displayed in Figs. 1 and 2, and in the Supplementary Information Files have been deposited in the Zenodo database under https://doi.org/10.5281/zenodo.5762083. The area-of-habitat loss data of each species generated in this study, displayed in Figs. 3 and 4, and Supplementary Fig. 6, are provided in the Supplementary Data files 7–10. The range map and habitat affiliation data for amphibians, reptiles and mammals, and the elevational range data of all taxonomic groups used in this study are available in the IUCN Red List of Threatened Species database under https://www.iucnredlist.org/. The range map data for birds used in this study are available in the BirdLife Bird species distribution maps of the world database and can be requested under http://datazone.birdlife.org/species/requestdis. The biome (broad ecosystem types) data used in this study to define area-of-habitats are available in the Ecoregions 2017 database under https://ecoregions.appspot.com/. The global elevation data used in this study are available in the GMTED2010 database under https://topotools.cr.usgs.gov/gmted_viewer/gmted2010_global_grids.php. The nitrogen application rates data used in this study are available in the Earthstat database under http://www.earthstat.org/nutrient-application-major-crops/. The gridded livestock information data used in this study are available in the FAO database under https://www.fao.org/livestock-systems/global-distributions/en/.

## Code availability

The R codes used to run the cSAR model and to calculate the native area-of-habitat loss of individual species are provided in the Supplementary Software file.

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

## Acknowledgements

We appreciate the help of Birdlife International for supplying data, and Matthew Forrest and Thomas Hickler from the Senckenberg Biodiversity and Climate Research Centre for supplying data and valuable advice for the HANPP calculations. The pictograms in maps were taken from www.phylopic.org. Funding was provided by the Vienna Science and Technology Fund (WWTF) through project No. ESR17-014 (P.S., S.D., F.E., G.B., T.K., K.H.E., H.H., C.P., S.M., F.K.), the German Federal Ministry of Education and Research within the framework of the TransRegBio project grant No. 031B0901A (T.K.), and the Austrian Science Fund (FWF) project No. P29130-G27 GELUC (H.H., K.H.E.).

## Author contributions

P.S., F.E., H.H., F.K. & S.D. designed the study. P.S. compiled species data, scaled intensity indices, and performed all biodiversity calculations. C.P., S.M., T.K. and K.H.E. compiled and calculated land use area and all HANPP related data. G.B. and T.K. compiled nitrogen application data. C.P. compiled gridded livestock and built-up area data. P.S., C.P. and J.W. created maps and figures. P.S. and S.D. wrote the text with contributions from all authors.

## Competing interests

The authors declare no competing interests.
