## [Peer Review File · Nature Communications]

Peer review comments, initial round review

Reviewer #1 (Remarks to the Author):

This work uses various global datasets to assess the relative effects of land use and land-use intensity on terrestrial vertebrate biodiversity. The effect of land use intensity is not often considered within global studies due to the coarse nature and correlations between available data. This work is therefore an important contribution into the assessment of the impacts of intensification on biodiversity across large scales. There is also an exploration of a simple analysis of potential impact of increasing intensification across current land use area.

I particularly appreciated the assessment of two sets of indicators of intensification, however, I would have liked to have seen a greater exploration into these two sets of results. At the moment there is no indication why these were assessed separately and then the results averaged. I assume it is because they are correlated.

Additionally, I would like to see a bit more information on the methods and datasets used within the main text. At the moment it is very challenging to get the gist of what has been done. This is also the case in places in the main methods section where there are a lot of references to papers without a brief description of what kind of data or output has been used from them. I have tried to give specific examples below.

Main text:

Line 43 – It would be beneficial to the reader to include a little more information on the “plot-scale data” mentioned in this line. At the moment this is very vague, and I think it would be useful to just name the study and/or type of data used i.e. estimates of impact derived from the PREDICTS database. If you are using the model coefficients of the model on the effect of land use on local biodiversity rather than the data itself, can this be described as “plot-scale” ? It would be the average response from a mixed effect model. As mentioned above, a little more transparency with what is being used would help with understanding what has been done.

Line 60 – how is extinction debt accounted for, I don’t fully understand how this has been incorporated and it is not mentioned in the methods.

Line 62 – Since the coefficients from the paper referenced here and the same broad land use categories have been used to determine the effect of intensification, is it not expected that results would be similar to this study? This is another point where clarification on what has been used for which part of the analysis could help with understanding, perhaps I have got the wrong end of the stick!

Line 105: I think “impeding” should be “impending” here.

Line 119-121 – This line isn’t too clear. What are the conceptual problems being highlighted here, there hasn’t been a mention of why including use-intensity is challenging. And again, clarification on what has been used from the 2014 PREDICTS study would be beneficial. Is the similarity to the other referenced papers because they have used the same model coefficients?

Figure 2 legend – Figure 8ED is not split by taxonomic group as stated in this figure legend.

Figure 1 – the authors might consider splitting out the maps based on the use-intensity results from the land use only results since they are discussed separately.

Methods:

Line 319 and 328: Is the reference raster just a 5x5 arm min grid or something else?

Line 377 – this is the first mention of PREDICTS and it is not clear why it is needed. This is an instance where it is not clear from earlier description in the methods and the main text which parts of which paper/datasets are being used.

Line 430 – why were the two approaches used and separated? Set 1 and set 2 could be better separated and described in this section. It is also not clear why the two sets were considered separately but then reported as a combined result.

Extended data figures:

Figure 3ED: not clear how the percentage of intensity is determined here.

Reviewer #2 (Remarks to the Author):

A review of "A relative of land conversion and land-use intensity on terrestrial vertebrate diversity."

This manuscript examines impending biodiversity loss from current land-use patterns, explicitly highlighting the role of land-use intensity. The work is certainly interesting, generally well done, and presented well (with a few exceptions, see below). There is no doubt that land-use intensification plays a role in shaping current terrestrial diversity, and that it's rarely assessed in a comprehensive way. Most due to a lack of data on the subject. Here, the authors present two sets of spatially explicit LU-intensity indicators, each combining different metrics, and state they will compare and combine their impacts on predicted species loss. However, what they seem to have done is to combine say estimates and only present and discuss their overall impacts. I would have like to see how both indices compare, are both Sets in agreement? What is Set 1 picking up that Set 2 may be neglecting and vice-versa? Which Set do the authors advise to be used in further work? The authors go into detail on how such sets were constructed in the methods, but there is no effort made to actually compare them, both in nature and their predictive capacity.

The narrative of the manuscript is, both LU change and LU intensity effects where and which species remain in a landscape, we therefore document how both have impacted terrestrial vertebrate diversity. However, the authors mostly present global average, which while not necessarily wrong, gives little indication of spatial variation of these impacts. While one could look at Figure 1, ED1, ED2 for some insights, I would have liked to see some regional breakdown (mean, CIs, max, min, etc) by administrative boundaries or/and bioclimatic boundaries. I suspect not all regions but specific regions, like Southeast Asia, will be disproportionately contributing to such global patterns. In addition, If the goal of the manuscript is to quantify both effects on impending species loss across the globe, I urge the authors to put more effort into communicating these results (perhaps re-thinking some of the figures – I found several hard to interpret) and what these results mean - How can we use these findings, and for what purpose can we use these findings? It's not fully clear what is the key messages and significance of this analysis. Statements such as "Our results suggest that LU-intensity at current levels contributes less than a third to both native species richness loss and AOH reductions" just add to the confusion by giving the impression that the role of LU-intensity is lower than expected. I will argue that LU-intensity being responsible for a third of the loss (even when most areas are under fairly low-intensity levels) is quite a high effect.

However, the whole LU-intensity scenario is problematic. I guess it is a nice theoretical exercise, but if one increases intensification levels to such scale, it is not surprising these effects will increase. This scenario carries some big assumptions, particularly that land conversion doesn't increase as well (and consequent the extent of each LU type). I fail to see how the authors can state that such impacts will excel conversion effects, without quantifying effects of raising land-use conversion as well (at least keeping with present-day LU conversion trends). One cannot realistically un-compelled one from the other. Still, if the author what to keep such scenario, please make it clear what this scenario accomplishes and how such results should be interpreted. In addition, it's well known that LU intensification is often coupled with specific landscapes changes (e.g. farmland abandonment of less productive lands). While I don't argue that the loss of such areas e.g. European cultural landscapes, will carry substantial biodiversity loss, land sparing can be quite beneficial in areas with no LU history, which accounts for a lot of the areas where more recent LU conversion is taking place. Discussing such nuances may enrich the overall manuscript. As of right now, the conclusion section is a bit too general (see above comment as well).

Finally, how dependent are species AOH patterns in Figure 2 from the species range size (i.e. number of cells they are present)? Species vulnerability to change is intrinsically linked to their range size, with small range species more likely to be vulnerable to LU-Conversion (and LU intensity). The authors show some of these links in Figure 3, but I wonder how depend are the patterns in Figure 2 are from this?

Data and methodology

I'm a bit confused about the categorization of the Land-uses, particularly how many were actually use. The authors state that they use 45 LU types, but then that they were aggregated in 6 broad

classes when actually carrying on the cSAR calculations. But later they say they calculated the cSAR with all 45 classes. Why aggregate? Where and until when did the 45 classes were used? The change between 6 classes and 45 classes seems to occur several times across the manuscript. This needs to be made clear.

The cSAR model is at its core a very simple model, with one key parameter: the affinity of species to each habitat: h . This parameter allows the model to account for the differential use of habitats by different species groups. Thus, the proportion of species disappearing/remaining in the landscape after land-use conversion is highly dependent on this parameter. As described by Pereira & Daily 2006 and Martins and Pereira 2017, affinity values can be derived in a few ways, but often represent a measure of the sensitivity of species to the full conversion of native habitat into the modified habitat. Here the authors propose their own way of calculating such affinities. While I applaud the effort the authors did in adapting say parameter to include LU-intensity levels, I have a few concerns on the steps they took that need to be clarified. How did the authors retrieve the intensity f functions (line 508-516)? How were the values EST_{int} and $EST_{minimal}$ determined here? The authors list two references (Newbold et al 2015 and the PREDICTS dataset). Did they use the values shown in Figure 1 from Newbold et 2015 or did they use the raw data of PREDICTS to estimate them themselves? If it's the former I don't seem to relate say information to the one presented in Figure ED6 (e.g. in Figure 1 of Newbold $builtup=urban$ is the LU type with the biggest SR differences between intensity levels). Furthermore, equation 3 in this section (please number the equations) needs some more explanation. What are the interpolated values and how do they yield an intercept and slope?

In terms of notation, what is the difference between f and f^{int} ? The subscript n is also used from the z parameter, but I don't think the authors have grid cell-specific z values right? Finally, while not influencing the calculations, it will be more comparable to the literature if the $1/z_n$ superscript is applied in equation 4 to both parameters (and removed from equation 1).

Finally, I wonder why the two lists of species (species affiliated and unaffiliated with each LU type) were not used from the beginning of the analysis? Again, one of the advantages of the cSAR approach over others is the model capacity to account for the differential use of habitats by different species functional groups (not just different taxon), so potentially one could assess the impact of LU-conversion and LU-intensity not only e.g. all birds, but specifically on birds with affinity to forest, birds with affinity to cropland, birds with multiple affinities, etc..) (e.g., Guilherme & Pereira 2013, Martins et al. 2020). I understand this analysis may be outside the scope of this study but can present intriguing avenues

Minor comments:

Line 51-52 and Figure 1 caption: Use the same way of referring to the land-use activities (i.e present-day or current).

Line 60-62: This is not clear.

Line 146: shouldn't be "species with largest AOH"?

Line 324-326: Fix numbering.

Line: 332-333: "Forth, we exclude species which were unaffiliated with that cell's pristine ecosystem" - not sure what the authors mean by this.

Line 335:356: Was this division carried out with the single purpose of dividing total cropland into what's actually cropland to what was plantation?

Line 408 – Fix reference.

Line 410-411: What does this mean?

Line 434-435: Where did these percentages come from?

Line 461:467 + Fig.ED6 – The scaling needs to be explained in more detail. What do the authors

mean with 'above which further intensification ceases to increase species richness loss'? (the same in Line 483). I found figure ED6 particularly hard to read.

Line 483: Please clarify.

Line 535: add (S^{lossint}) to the subsection header

Line 541: I think this is the first time S^{loss} is mention. Is this present-day land conversion loss? Please define.

Line 551: Provide the exact name of the section instead.

Line 570-577: This seems to be critical for Figure 2 and needs to be better explained.

Figure ED5 – I find this figure more interesting than most, as it provides a clear message.

Reviewer #3 (Remarks to the Author):

This is an outstanding paper that examines with unprecedented detail the impacts of land-use intensity on biodiversity. The authors achieve that by improving the countryside SAR to account for land-use intensity and by bringing together state of the art datasets about land use cover and land use intensity. I think the approach the authors followed to scale the impacts of land-use intensity by land-use type and incorporate it in the cSAR is brilliant. I only have two major and a few very minor comments for the authors, which I hope can be helpful.

General comments

1. The scenario of all land-use being converted to high intensity does not seem to be very realistic. Therefore, I was a bit struck that in the abstract they highlight that result, when there are so many other interesting things to highlight instead (e.g. the impacts of low land-use intensity areas just by the magnitude of their size, l.102-103).

2. The authors report global averages without wilderness landscapes. These make the values of species loss significantly higher and hard to compare with other studies where truly global averages are calculated (typically excluding ice-dominated landscapes). I think it would be good if the authors would report both their wilderness free average and the total average.

Specific comments

l.115-117. I believe the statement here is too harsh on studies that do not account for LU intensity. Something along the lines of "may underestimate" would perhaps be in order. I think we should not completely throw away studies that just do land use conversion, as they provide important estimates to be compared with those that include LU intensity.

l.125-127 I got a bit lost here and I am not sure how the cSAR models masks species that had immigrated into focal landscapes - this seems a bit speculative.

l.140-142. I am not sure about this focus on the confidence limits of the AOH. Why not discuss only the median values?

l.151-153. You may want to consider to cite Staude et al (2020), GEB, DOI: 10.1111/geb.13003. They looked at the relationship between amount of habitat loss in a grid cell and the probability of extinction, and they found narrow range species are subject to a double jeopardy: they are both more vulnerable to have their small range wiped out (contrasting a bit with the results of your Figure 3, but in reality they did not test this explicitly) and are more vulnerable in terms of "landscape"/grid-cell level extinction probability (they do show this clearly).

l. 394-348. I think a table summarising all these options in indicator set 1 and 2 would be helpful.

l.413 Any reason for 1/2, 1/3 numbers. or they are just educated guesses?

Henrique M. Pereira

Reviewer #1 (Remarks to the Author):

This work uses various global datasets to assess the relative effects of land use and land-use intensity on terrestrial vertebrate biodiversity. The effect of land use intensity is not often considered within global studies due to the coarse nature and correlations between available data. This work is therefore an important contribution into the assessment of the impacts of intensification on biodiversity across large scales. There is also an exploration of a simple analysis of potential impact of increasing intensification across current land use area.

I particularly appreciated the assessment of two sets of indicators of intensification, however, I would have liked to have seen a greater exploration into these two sets of results. At the moment there is no indication why these were assessed separately and then the results averaged. I assume it is because they are correlated.

We are thankful to the reviewer for valuing our efforts to include different intensity indicators in the assessment. We agree that it is worthwhile to elaborate more on the two LU-intensity indicator sets. We now keep the two sets separate in most cases, and included a more thorough discussion on this matter (including a reference to SI Tables 1 and 2, and new Fig. 2) in the results (starts l 129) and discussion (starts l 230) sections. See also comments to reviewer 2 who made a similar suggestion.

Additionally, I would like to see a bit more information on the methods and datasets used within the main text. At the moment it is very challenging to get the gist of what has been done. This is also the case in places in the main methods section where there are a lot of references to papers without a brief description of what kind of data or output has been used from them. I have tried to give specific examples below.

Thanks to the reviewer for pointing this issue out. We have changed the text accordingly and included more detailed method descriptions throughout the main text and the updated and expanded methods section. For details see comments below.

Main text:

Line 43 – It would be beneficial to the reader to include a little more information on the “plot-scale data” mentioned in this line. At the moment this is very vague, and I think it would be useful to just name the study and/or type of data used i.e. estimates of impact derived from the PREDICTS database. If you are using the model coefficients of the model on the effect of land use on local biodiversity rather than the data itself, can this be described as “plot-scale” ? It would be the average response from a mixed effect model. As mentioned above, a little more transparency with what is being used would help with understanding what has been done.

We have changed the sentence (now in l 59) to: “This approach is based on linking model coefficients from ref¹, who analyse local assemblage data from the PREDICTS database², with spatially explicit LU-intensity indicators [...].” See also the corresponding changes in the methods section (see comments below).

Line 60 – how is extinction debt accounted for, I don't fully understand how this has been incorporated and it is not mentioned in the methods.

Extinction debt is not explicitly accounted for in our calculations. We just wanted to express here that the cSAR model can only predict how many species will find suitable habitat in an area changed by land use, but not how long it will take until populations of the species that do not find suitable area there anymore will actually go extinct – it is a prediction of an eventual equilibrium stage and not of the dynamics during the relaxation time. To avoid confusion, we removed this statement from the results section and moved it to the discussion instead, with further explanation. See l. 200: “The species loss metrics calculated here refer to an eventual equilibrium situation between the habitat mosaic created by LU and the number of native species able to persist in this mosaic, i.e. to a situation in which all extinction debt has been paid off³. However, remnant populations may persist for long times, and individuals from neighbouring source populations may help maintaining vital populations in an otherwise unsuitable landscape. Also, our results refer to the impending loss of species adapted to the pristine ecosystems of these landscapes only⁴, while they do not account for possible immigration of novel, non-native species adapted to the landscapes created by LU⁵ from outside a focal region or landscape. In summary, net realized species loss, as measured through e.g. field surveys, may be lower than the predictions of our model”.

Line 62 – Since the coefficients from the paper referenced here and the same broad land use categories have been used to determine the effect of intensification, is it not expected that results would be similar to this study? This is another point where clarification on what has been used for which part of the analysis could help with understanding, perhaps I have got the wrong end of the stick!

We agree in principle, but emphasize that the portion of total species losses where these coefficients were used only contribute about a quarter to the estimated species loss (= effect of LU-intensity), while three quarters come from land conversion, which has been quantified in a completely different way. To avoid confusion, we moved this statement to the discussion and expanded it, see the third paragraph of the discussion (starting in l. 220): “In contrast to the cSAR model, species lists from local assemblages do include, to an unknown extent, species present because of unpaid extinction debt or immigration in response to human usage. Despite these differences, the global average of species loss calculated here is similar in magnitude to the one reported by a recent meta-analysis of such local assemblage data¹. This similarity may in part be due to the fact that we used model coefficients from this study to quantify the effect of LU-intensity in our cSAR model (Methods). Nevertheless, the bulk of our calculations was based on data sources other than ref¹ and the results are, hence, independent. Further, LU-intensity only contributes a quarter to our results while the remainder is driven by land conversion which has been parameterized in a completely different way. From the agreement of the two studies we tentatively conclude that a value of c. 15% can be considered a plausible, robust estimate of the average magnitude of LU-driven species loss in current terrestrial environments”.

Line 105: I think “impeding” should be “impending” here.

That's correct, fixed!

Line 119-121 – This line isn't too clear. What are the conceptual problems being highlighted here, there hasn't been a mention of why including use-intensity is challenging. And again, clarification on what has been used from the 2014 PREDICTS study would be beneficial. Is the similarity to the other referenced papers because they have used the same model coefficients?

Thanks to the reviewer for this remark. We revised the section for more clarity. We see two main problems – first, there are few biodiversity data that allow exploring the relationship between species richness and land-use intensity at the landscape scale, e.g. 5 x 5 min as used in our calculations. Second, land-use intensity is a complex, multi-faceted phenomenon with many possible direct and indirect effects on biodiversity (see ref⁶ for a recent paper on this issue), and it is hence not a priori clear how to represent LU-intensity appropriately in biodiversity modelling. We now expand on these issues in the new Discussion section starting in l. 231. Regarding the question of similarity, please see our response to the previous comment of this referee.

Figure 2 legend – Figure 8ED is not split by taxonomic group as stated in this figure legend.

Thanks for pointing this out. The figure you are referring to is now Fig. 3, and in the caption we meant to refer to Fig. ED7 instead of ED8. This is fixed now.

Figure 1 – the authors might consider splitting out the maps based on the use-intensity results from the land use only results since they are discussed separately.

We are happy about this suggestion and changed the figure such, that only the current total species loss is shown. We also refer to the new Fig. 2 where land-use intensity alone is shown, together with the comparison between the two sets. As we don't focus on the counterfactual scenario so much anymore, we removed the maps of that altogether to avoid putting too much weight on it.

Methods:

Line 319 and 328: Is the reference raster just a 5x5 arcmin grid or something else?

Added in l 293: “[...] reference 5 x 5 arcmin grid [...]”.

Line 377 – this is the first mention of PREDICTS and it is not clear why it is needed. This is an instance where it is not clear from earlier description in the methods and the main text which parts of which paper/datasets are being used.

The instance you are referring to is now in l 377, and we added the following reference to a later section of the text: “(needed for quantifying LU-intensity effects, see below in section “Affinity parameter” for details)”.

In the named section, l 515 has been expanded to: “These functions were derived from estimates of linear mixed effects models published in a recent analysis of fractional species richness declines in local assemblages. The authors used the PREDICTS database and distinguished four broad LU-

types and three categorical intensity levels (*Minimal, Light, Intense*)^{1,2}. The four broad LU-types from this study were assigned to the six broad LU types we used in our aggregation (see SI Tab.2). Note that while cSAR calculations were performed on LU-maps that distinguished all 45 LU-types, the intensity-driven loss functions were identical to all types belonging to one of the four categories distinguished in PREDICTS, e.g. for all 30 annual crop types assigned to the cropland category or for all 11 permanent crop types plus timber plantations assigned to the plantation category (see SI Tab. 2 for details)".

Line 430 – why were the two approaches used and separated? Set 1 and set 2 could be better separated and described in this section. It is also not clear why the two sets were considered separately but then reported as a combined result.

We now included a comparison of the two intensity indicator sets in the results (starts l 130, new Figs. 2) and discussion section (starts l 231), and added the following sentence in l 400: "We used two indicator sets to account for the multidimensional nature of LU-intensity^{6,7} and to include a wide range of available data products".

Extended data figures:

Figure 3ED: not clear how the percentage of intensity is determined here.

Thanks for pointing this mistake out. We changed the legend of the colour bar to the proportion scale, accordingly.

Reviewer #2 (Remarks to the Author):

A review of "A relative of land conversion and land-use intensity on terrestrial vertebrate diversity.

This manuscript examines impending biodiversity loss from current land-use patterns, explicitly highlighting the role of land-use intensity. The work is certainly interesting, generally well done, and presented well (with a few exceptions, see below). There is no doubt that land-use intensification plays a role in shaping current terrestrial diversity, and that it's rarely assessed in a comprehensive way. Most due to a lack of data on the subject. Here, the authors present two sets of spatially explicit LU-intensity indicators, each combining different metrics, and state they will compare and combine their impacts on predicted species loss. However, what they seem to have done is to combine say estimates and only present and discuss their overall impacts. I would have like to see how both indices compare, are both Sets in agreement? What is Set 1 picking up that Set 2 may be neglecting and vice-versa? Which Set do the authors advise to be used in further work? The authors go into detail on how such sets were constructed in the methods, but there is no effort made to actually compare them, both in nature and their predictive capacity.

We initially combined the two intensity indicator sets to avoid overwhelming the reading with too many numerical results. However, and also to comply with comments made by reviewer 1, we agree that this is an important issue of our study that deserves more attention. Hence, we included an explicit comparison of the results achieved with the two sets in the results (starts l

130, new Figs. 2) and discussion sections (starts l 231). See also responses to comments of reviewer 1.

The line of our revision is now to explicitly acknowledge the multidimensional nature of land-use intensity (see for instance refs^{6,7}). As pointed out in these references, the causal links between land management and biodiversity loss are manifold and potentially complex. As a consequence, there is no generally best indicator of LU-intensity for biodiversity research. Furthermore, data uncertainty related to any land-use information, including land-use intensity, is large. Therefore, the use of more than one indicator, ideally representing different dimensions of LU-intensity such as input, output and system dimensions⁸ appears advisable in biodiversity research. The two sets we use in this study reflect these different dimensions as described in the Introduction (starting in line 61).

We emphasize that the quality and accuracy of global land-use intensity indicator maps differs among indicators and it is generally far from perfect, especially at the spatial scale of 5 x 5 arcmin^{6,9}. Against this background it is re-assuring that the two sets we used produced similar results in overall terms, but with some differences at the regional levels. It is not straightforward to decide which of the two sets would be superior over the other by delivering more plausible results. Therefore, we revised the text and now explicitly address - wherever meaningful - the two results separately and in comparison to each other, rather than merging them. This allows us to elaborate on differences (e.g. Fig2), but also to assess the robustness of the overall results we obtain. We interpret the fact that, at the global level, both datasets result in similar result as an indication that the LU-intensity effect we quantify is relatively robust.

The narrative of the manuscript is, both LU change and LU intensity effects where and which species remain in a landscape, we therefore document how both have impacted terrestrial vertebrate diversity. However, the authors mostly present global average, which while not necessarily wrong, gives little indication of spatial variation of these impacts. While one could look at Figure 1, ED1, ED2 for some insights, I would have liked to see some regional breakdown (mean, CIs, max, min, etc) by administrative boundaries or/and bioclimatic boundaries. I suspect not all regions but specific regions, like Southeast Asia, will be disproportionally contributing to such global patterns.

We thank the reviewer for this idea, we calculated national and biome-wide average species losses and put them in the new SI Tables 5 (national) and 6 (biomes). We chose the Ecoregion 2017 biomes as ecologically relevant regions, as we have also used them to define the area-of-habitat of native species. We included statements on species richness losses in biomes throughout the results (l 88 and l 116) section. For the sake of readability, we are not further expanding on the results of national averages, however, include them in the SI for the readers to access.

In addition, If the goal of the manuscript is to quantify both effects on impending species loss across the globe, I urge the authors to put more effort into communicating these results (perhaps re-thinking some of the figures – I found several hard to interpret) and what these results mean - How can we use these findings, and for what purpose can we use these findings? It's not fully clear what is the key messages and significance of this analysis. Statements such as "Our results suggest that LU-intensity at current levels contributes less than a third to both native species richness loss and AOH reductions" just add to the confusion by giving the impression that the role of LU-intensity is

lower than expected. I will argue that LU-intensity being responsible for a third of the loss (even when most areas are under fairly low-intensity levels) is quite a high effect.

We thank the reviewer for this assertion. Indeed, the main aim of the paper is disentangling the effects of land conversion and land-use intensity on biodiversity loss. To make that clearer, we reformulated the Abstract, changed Fig. 1 and introduced the new Fig. 2 which specifically compares intensity effects achieved with the two different LU-intensity-indicator sets, and highlighted the relative contributions of land-use intensity at the beginning of the (new) Discussion section. We also agree that the ca. one quarter contribution we estimate on the part of land-use intensity is substantial and reformulated the pertinent passages such as Abstract and Discussion accordingly, emphasizing that omitting the land-use intensity effects in biodiversity studies yields incomplete results and potentially skewed interpretations.

Concerning the “use” of the study we think there are two aspects: first, a thorough quantitative estimate of the relative contributions of land conversion and land-use intensity to biodiversity loss, such as the one we present, is per se novel and of interest; and, second, highlighting the considerable impact land-use intensity already has, and the scale to which this impact could increase, represents a contribution to the general discussion about how to best reconcile challenges of land-product security and biodiversity conservation. It highlights, for example, that increasing food production by intensification without further land conversion still bears considerable risks for biodiversity, and that, hence, the way land-use management regimes are intensified is of crucial importance. We emphasize this aspect throughout the Discussion section.

However, the whole LU-intensity scenario is problematic. I guess it is a nice theoretical exercise, but if one increases intensification levels to such scale, it is not surprising these effects will increase. This scenario carries some big assumptions, particularly that land conversion doesn't increase as well (and consequent the extent of each LU type). I fail to see how the authors can state that such impacts will excel conversion effects, without quantifying effects of raising land-use conversion as well (at least keeping with present-day LU conversion trends). One cannot realistically un-compelled one from the other. Still, if the author what to keep such scenario, please make it clear what this scenario accomplishes and how such results should be interpreted. In addition, it's well known that LU intensification is often coupled with specific landscapes changes (e.g. farmland abandonment of less productive lands). While I don't argue that the loss of such areas e.g. European cultural landscapes, will carry substantial biodiversity loss, land sparing can be quite beneficial in areas with no LU history, which accounts for a lot of the areas where more recent LU conversion is taking place. Discussing such nuances may enrich the overall manuscript. As of right now, the conclusion section is a bit too general (see above comment as well).

We fully agree that the scenario we use here is an oversimplified theoretical one that was not intended as prognosis or realistic scenario. Exploring full-fledged realistic scenarios of future land use that account for both conversion and intensification is a complex undertaking beyond the scope of this paper (see e.g. ref¹⁰). The scenario's only function here is to highlight the scale of further biodiversity loss that could be triggered by intensification alone. We therefore decided to keep this element in the paper, but now make the explorative nature of this “thought experiment” explicit and clear already at the very beginning of the Discussion section (and refer to it as “counterfactual” throughout the manuscript). We also agree that warning against the biodiversity effects of further intensification does not make the case for further land conversions or diminishes

the value of halting conversion has for biodiversity, especially in regions such as remaining primary rain forests. We have now added this aspect to the final part of the Discussion.

Finally, how dependent are species AOH patterns in Figure 2 from the species range size (i.e. number of cells they are present)? Species vulnerability to change is intrinsically linked to their range size, with small range species more likely to be vulnerable to LU-Conversion (and LU intensity). The authors show some of these links in Figure 3, but I wonder how dependent are the patterns in Figure 2 from this?

The Figures 2 and 3 the referee refers to are now Figs. 3 and 4. As we see it, Fig. 4 shows just what the referee is describing, and we are hence a bit unsure how to respond to this comment. See also statements in l 164 and the paragraph in the discussion starting in l 248, which both refer to these issues.

Data and methodology

I'm a bit confused about the categorization of the Land-uses, particularly how many were actually used. The authors state that they use 45 LU types, but then that they were aggregated in 6 broad classes when actually carrying on the cSAR calculations. But later they say they calculated the cSAR with all 45 classes. Why aggregate? Where and until when did the 45 classes were used? The change between 6 classes and 45 classes seems to occur several times across the manuscript. This needs to be made clear.

We apologize for this confusion. We have revised the relevant sections for clarity. In short, we used the cSAR model to calculate species loss based on a map distinguishing 45 land-use types. However, to combine information from different data sources, e.g. IUCN habitat information of species, land-use intensity indicator maps or estimates of land-use intensity effects on biodiversity (form ref¹) we had to produce an assignment table aligning the different categorizations used in these sources with each other. This was only possible by combining the 45 types into the broader categories used by these other sources. As a consequence, the spatial configuration of LU-types is resolved to the level of 45 categories, but the information on e.g. the intensity of usage is independent of which type of e.g. crop is cultivated in a particular 5 x 5 arcmin cell (as the intensity indicator only distinguishes the broad cropland category as such).

In the manuscript we tried to clarify this issue by adding text where needed. The key information is now contained in the following four statements:

L 298: "For presentation of results, we aggregated the calculated effects of the 45 LU types into those of six broad LU types (cropland (30 annual crop types); pastures (non-grassland converted to grassland); grazing land (natural/ near-natural areas with livestock grazing); builtup (sealed areas); plantations (11 permanent crop types plus timber plantations), and forests (natural/ near-natural forest under forestry); see SI Table 2 for details)" .

L 392: "As stated above, these aggregated broad LU types were needed to align the different LU categorizations used in the different data sources with each other. The effects on biodiversity were then calculated on each of the 45 LU types and afterwards aggregated to the six broad LU types to give a better overview".

L 522: “The four broad LU-types from this study were assigned to the six broad LU types we used in our aggregation (see SI Tab.2). Note that while cSAR calculations were performed on LU-maps that distinguished all 45 LU-types, the intensity-driven loss functions were identical to all types belonging to one of the four categories distinguished in PREDICTS, e.g. for all 30 annual crop types assigned to the cropland category or for all 11 permanent crop types plus timber plantations assigned to the plantation category (see SI Tab. 2 for details)”.

L 603: “All cSAR model calculations were based on global land use maps that distinguish 45 LU-types as described above. For the sake of simplicity, we present results aggregated to the six broad LU-types cropland, pastures, natural grazing land, built-up, plantations, and forests (natural/ near-natural forest under forestry; see SI Tab. 2)”.

The cSAR model is at its core a very simple model, with one key parameter: the affinity of species to each habitat: h . This parameter allows the model to account for the differential use of habitats by different species groups. Thus, the proportion of species disappearing/remaining in the landscape after land-use conversion is highly dependent on this parameter. As described by Pereira & Daily 2006 and Martins and Pereira 2017, affinity values can be derived in a few ways, but often represent a measure of the sensitivity of species to the full conversion of native habitat into the modified habitat. Here the authors propose their own way of calculating such affinities. While I applaud the effort the authors did in adapting say parameter to include LU-intensity levels, I have a few concerns on the steps they took that need to be clarified. How did the authors retrieve the intensity functions (line 508-516)? How were the values EST_{int} and $EST_{minimal}$ determined here? The authors list two references (Newbold et al 2015 and the PREDICTS dataset). Did they use the values shown in Figure 1 from Newbold et 2015 or did they use the raw data of PREDICTS to estimate them themselves? If it's the former I don't seem to relate say information to the one presented in Figure ED6 (e.g. in Figure 1 of Newbold builtup=urban is the LU type with the biggest SR differences between intensity levels). Furthermore, equation 3 in this section (please number the equations) needs some more explanation. What are the interpolated values and how do they yield an intercept and slope?

We apologize for the confusion caused by the description of our method to arrive at the functions and clarified these concerns in the following way.

EST_{int} and EST_{minimal} were, as correctly assumed, directly taken from Newbold et al. 2015, and we clarified that point in I 519 (see also comments to reviewer 1): “These functions were derived from estimates of linear mixed effects models published in a recent analysis of fractional species richness declines in local assemblages. The authors used the PREDICTS database and distinguished four broad LU-types and three categorical intensity levels (*Minimal, Light, Intense*)”.

This information obviously did not fit with the former Fig. ED6, because unfortunately the naming of the panels was scrambled. We have corrected this now, see the new Fig. ED6, where the patterns from Figure 1 in ref¹ (i.e. Newbold et al. 2015) are now recognizable.

For clarification on how we yielded the intercepts and slopes in eq. 4 (referred to by the referee as eq. 3), we added the following statement in I 538: “We then placed these $s_{n,b}$ values in the respective coordinate system (see Fig. ED6) and linearly interpolated between the values to yield an intercept and slope for each interpolation step. This procedure allowed us to calculate as (eq. 4)”.

Besides these changes, we numbered all equations throughout the methods section.

In terms of notation, what is the difference between f and f^{int} ? The subscript n is also used from the z parameter, but I don't think the authors have grid cell-specific z values right? Finally, while not influencing the calculations, it will be more comparable to the literature if the $1/z^n$ superscript is applied in equation 4 to both parameters (and removed from equation 1).

To avoid confusions, we renamed f^{int} to s (in eq. 3 and the corresponding text).

The z -values are for three landscape types taken from Drakare et al. 2006, and each grid cell is assigned to one of the three types. So technically speaking, while there are only three distinct z -values, each cell has nevertheless its own specific value (= one of these three).

We further changed the superscripts as suggested.

Finally, I wonder why the two lists of species (species affiliated and unaffiliated with each LU type) were not used from the beginning of the analysis? Again, one of the advantages of the cSAR approach over others is the model capacity to account for the differential use of habitats by different species functional groups (not just different taxon), so potentially one could assess the impact of LU-conversion and LU-intensity not only e.g. all birds, but specifically on birds with affinity to forest, birds with affinity to cropland, birds with multiple affinities, etc..) (e.g., Guilherme & Pereira 2013, Martins et al. 2020). I understand this analysis may be outside the scope of this study but can present intriguing avenues

This is an interesting suggestion and we agree that a breakdown into species groups could yield interesting results. However, as said by the referee, the manuscript already contains quite a number of detailed results and we hence prefer to reserve such further breakdown of calculations to an additional study.

Minor comments:

Line 51-52 and Figure 1 caption: Use the same way of referring to the land-use activities (i.e present-day or current).

Changed "present-day" to "current".

Line 60-62: This is not clear.

The sentence (now l 77-80) has been changed substantially, also in response to other comments.

Line 146: shouldn't be "species with largest AOH"?

This is now in the new discussion section (l 248) and we specifically want to highlight here that species with very small AOH are a bit less affected than the ones with small to middle AOH. This

statement is now embedded in a whole new paragraph and we hope the message now comes out clearly.

Line 324-326: Fix numbering.

Done.

Line: 332-333: “Forth, we exclude species which were unaffiliated with that cell’s pristine ecosystem” - not sure what the authors mean by this.

We changed the sentence to: “Fourth, we removed species from this cell’s species list which, according to the IUCN, are not affiliated with that cell’s pristine ecosystem” (I 334).

Line 335:356: Was this division carried out with the single purpose of dividing total cropland into what’s actually cropland to what was plantation?

Yes, that’s correct. To comply with the IUCN habitat classification scheme, we needed the division of cropland in general and distribute some crops to plantations. In I. 355, we added the pre-sentence “To comply with the IUCN habitats classification scheme [...]”.

Line 408 – Fix reference.

This revision of this reference (I 413) is now under evaluation. We hope the paper will be accepted until this one is in the proof stage and will fix the reference then.

Line 410-411: What does this mean?

This is a technical detail needed to configure the LPJ-GUESS model and is intended to help people who want to repeat the exact steps done.

Line 434-435: Where did these percentages come from?

We added in the previous sentence (I 440): “[...] as reported in ref¹¹”.

Line 461:467 + Fig.ED6 – The scaling needs to be explained in more detail. What do the authors mean with ‘above which further intensification ceases to increase species richness loss’? (the same in Line 483). I found figure ED6 particularly hard to read.

We agree that a reference to Fig. ED6 at this place in the manuscript is a bit too early and removed it. How exactly the scaling has been done is explained in the paragraphs following this one (starting in I 473).

In addition, we added the following information in I 468: “[...] above which an increase of intensity causes no further increase of species loss”.

Line 483: Please clarify.

We added in l 488: “[...] i.e. after which an increase of N application rates causes no further increase in species loss [...]”.

Line 535: add ($S^{lossint}$) to the subsection header

Done.

Line 541: I think this is the first time S^{loss} is mention. Is this present-day land conversion loss? Please define.

We added “[...] and $S_{g,n}^{loss}$ from eq. 1” in l 564.

Line 551: Provide the exact name of the section instead.

We meant the section “Species loss caused by LU-intensity” and added that in l.571.

Line 570-577: This seems to be critical for Figure 2 and needs to be better explained.

It is correct that this is defining parts of the methods used to create the results presented in former Fig. 2 (which is now Fig. 3). However, maybe there is a misunderstanding on the complexity of this specific issue. It is really just about randomly choosing the order of LU types to be drawn from and added that in l 591: “[...] i.e. the order of LU types”.

Figure ED5 – I find this figure more interesting than most, as it provides a clear message.

Thanks for supporting this figure! However, as the focus of the manuscript has now changed somewhat (more towards the effects of land-use intensity), we now don’t need this figure anymore and removed it from the manuscript altogether.

Reviewer #3 (Remarks to the Author):

This is an outstanding paper that examines with unprecedented detail the impacts of land-use intensity on biodiversity. The authors achieve that by improving the countryside SAR to account for land-use intensity and by bringing together state of the art datasets about land use cover and land use intensity. I think the approach the authors followed to scale the impacts of land-use intensity by land-use type and incorporate it in the cSAR is brilliant. I only have two major and a few very minor comments for the authors, which I hope can be helpful.

We fully appreciate these kind words!

General comments

1. The scenario of all land-use being converted to high intensity does not seem to be very realistic. Therefore, I was a bit struck that in the abstract they highlight that result, when there are so many other interesting things to highlight instead (e.g. the impacts of low land-use intensity areas just by the magnitude of their size, l.102-103).

Thank to the reviewer for his frank evaluation of our intensification scenario. We fully agree it is unrealistic and it was never thought to be, and we refer to it as “counterfactual” to make this clear throughout the text. Its function in the text is to highlight the magnitude of further threat to biodiversity that could be caused by intensification alone. We agree, however, that it is of too little overall interest and thus removed it from the Abstract. We now highlight, as suggested, that the calculated intensity effect is one quarter, even though large parts of the globe are still used at low intensity. We also clearly emphasize that the scenario is not meant to be realistic but rather an exploration of upper boundary conditions at the beginning of the now revised Discussion section.

2. The authors report global averages without wilderness landscapes. These make the values of species loss significantly higher and hard to compare with other studies where truly global averages are calculated (typically excluding ice-dominated landscapes). I think it would be good if the authors would report both their wilderness free average and the total average.

We have included the average species loss rate calculated including wilderness landscapes in the text (l. 82). To avoid overloading the text with numbers, we report further more detailed results of calculations that include wilderness in SI Tab. 1.

Specific comments

l.115-117. I believe the statement here is too harsh on studies that do not account for LU intensity. Something along the lines of "may underestimate" would perhaps be in order. I think we should not completely throw away studies that just do land use conversion, as they provide important estimates to be compared with those that include LU intensity.

As we had to restructure the article in accordance with journal style the part on the intensification scenario was split among the new Results and Discussion sections and the commented formulation has been deleted. In general, we however agree that there is high value in conversion-only-based studies, but also that results obtained by such approaches need to be precise in the interpretation of their results and avoid, e.g. too general conclusion about the “land-use effects on biodiversity”, if only land conversion was studied.

l.125-127 I got a bit lost here and I am not sure how the cSAR models mask species that had immigrated into focal landscapes - this seems a bit speculative.

We have re-written this part, which is now in the discussion, l 212: “[...] the focus on pristine species pools and their depletion excludes the fate of species that had immigrated into a focal landscape in response to historical LU centuries or even millennia ago. These species are often

considered native today even if they were not present prior to the historical introduction of LU in a given landscape”.

I.140-142. I am not sure about this focus on the confidence limits of the AOH. Why not discuss only the median values?

We interpret species at the risk of global extinctions as the ones having their upper confidence limit as 100% AOH loss (instead of using medians or means) because these are the ones where total AOH loss is within the range of outcomes based on our random draw approach. We still think this is an arguable interpretation. Concentrating on medians would, in our opinion, result in underestimation, especially as small-range species often have wide confidence intervals, for probabilistic reasons, and medians therefore rarely reach levels close to 100%.

I.151-153. You may want to consider to cite Staude et al (2020), GEB, DOI: 10.1111/geb.13003. They looked at the relationship between amount of habitat loss in a grid cell and the probability of extinction, and they found narrow range species are subject to a double jeopardy: they are both more vulnerable to have their small range wiped out (contrasting a bit with the results of your Figure 3, but in reality they did not test this explicitly) and are more vulnerable in terms of "landscape"/grid-cell level extinction probability (they do show this clearly).

We thank the reviewer for pointing out this very useful publication which we included in the new discussion section in I 248 (ref 31 in the manuscript), embedded in the following statement: “We show that the risk of global extinction caused by LU is highest for geographically restricted species. Both the mean calculated AOH losses, as well as their upper confidence limits (and the confidence intervals’ widths) are higher for species with smaller native AOH (Fig. 4). A recent study found that grid-cell level persistence probabilities are lower for small-ranged species¹², i.e. that those species are more likely to disappear from a landscape in response to LU change than wide-ranged ones. This puts species with small native AOH under double jeopardy, as they (a) have higher landscape level extinction probabilities and are simultaneously (b) more likely to lose their entire AOH and go globally extinct. However, in our simulations, predicted AOH loss peaks for species with small native AOH, but levels out or decreases again for species with smallest AOH (Fig. 4). This may indicate that the distribution of the most narrowly distributed endemics is not linear^{13”}.

I. 394-348. I think a table summarising all these options in indicator set 1 and 2 would be helpful.

We have included an overview of the construction (data + short summary) of intensity indicator sets in SI Tab. 2.

I.413 Any reason for 1/2, 1/3 numbers. or they are just educated guesses?

No, these numbers were taken from the literature. We added in I 423: “[...] which was defined as 1/3 of the potential vegetation in ref^{14”}.

Henrique M. Pereira

References used in reviewer comments

1. Newbold, T. *et al.* Global effects of land use on local terrestrial biodiversity. *Nature* **520**, 45–50 (2015).
2. Hudson, L. N. *et al.* The PREDICTS database: A global database of how local terrestrial biodiversity responds to human impacts. *Ecology and Evolution* **4**, 4701–4735 (2014).
3. Tilman, D., May, R. M., Lehman, C. L. & Nowak, M. A. Habitat destruction and the extinction debt. *Nature* (1994) doi:10.1038/371065a0.
4. Dinerstein, E. *et al.* An Ecoregion-Based Approach to Protecting Half the Terrestrial Realm. *BioScience* **67**, 534–545 (2017).
5. Jackson, S. T. & Sax, D. F. Balancing biodiversity in a changing environment: extinction debt, immigration credit and species turnover. *Trends in Ecology and Evolution* (2010) doi:10.1016/j.tree.2009.10.001.
6. Dullinger, I. *et al.* Biodiversity models need to represent land-use intensity more comprehensively. *Global Ecology and Biogeography* geb.13289 (2021) doi:10.1111/geb.13289.
7. Kuemmerle, T. *et al.* Challenges and opportunities in mapping land use intensity globally. *Current Opinion in Environmental Sustainability* **5**, 484–493 (2013).
8. Erb, K. H. *et al.* A conceptual framework for analysing and measuring land-use intensity. *Current Opinion in Environmental Sustainability* **5**, 464–470 (2013).
9. Verburg, P. H. *et al.* Beyond land cover change: towards a new generation of land use models. *Current Opinion in Environmental Sustainability* **38**, (2019).
10. Erb, K.-H. *et al.* Exploring the biophysical option space for feeding the world without deforestation. *Nature Communications* **7**, 11382 (2016).
11. Krausmann, F. *et al.* Global human appropriation of net primary production doubled in the 20th century. *Proceedings of the National Academy of Sciences of the United States of America* (2013) doi:10.1073/pnas.1211349110.
12. Staude, I. R., Navarro, L. M. & Pereira, H. M. Range size predicts the risk of local extinction from habitat loss. *Global Ecology and Biogeography* **29**, 16–25 (2020).
13. Maxwell, S. L. *et al.* Area-based conservation in the twenty-first century. *Nature* **586**, 217–227 (2020).
14. Haberl, H. *et al.* Quantifying and mapping the human appropriation of net primary production in earth's terrestrial ecosystems. *Proceedings of the National Academy of Sciences of the United States of America* **104**, 12942–12947 (2007).

Peer review comments, further round review

Reviewer #1 (Remarks to the Author):

The authors have made a number of changes to the manuscript that help to clarify the methods, and now also highlight the two intensity indicators separately and compare their results in the main text and discussion. This helps to show the more novel element of the work more prominently. Just a thought, but would the authors be able to provide usable maps of their two use intensity indicators, perhaps as a data product, so that they can be used in the future by others? I know I would be able to find a use for them!

Abstract:

Line 29: Line starting "Land-use intensity contributes...". It feels like this sentence is stating 2 unrelated elements. Consider editing. I think removing the "although large areas of the globe are still used with low intensity" will help.

Mentioning the two sets of intensity indicators being tested and the data they focus on would help to present the novelty of this piece in the abstract. At the moment, very little of the abstract focuses on the intensity part.

Introduction:

The introduction, specifically paragraphs 1 and 2, does very little to set the scene before diving straight into the description of the main methods. I think it could be beneficial to better outline why including intensity is so important, why only coarse measures have been used so far, and the fact that intensity can relate to so many variables. Some of these points are interspersed into the third paragraph so could be incorporated earlier.

There is very little in terms of hypotheses or questions that you are trying to answer. I think that framing in this way would help with the structure. You state that you will "fill these gaps" but there is little linkage between the gaps, the methods and the results.

Line 43: Another global study that includes use intensity is the Newbold et al paper that you use in the methods. This could be added as another reference here along with ref 10. It is still a coarse measure of intensity.

Line 67: remove misplaced "is"

Line 67: around here, it could be useful to add something stating if/when averages across these two sets are considered.

There is no statement regarding the comparison of the two indicators or what differences might be expected. This would help set up the discussion of regional differences that appear later.

Results:

Line 90: I think the land use types can be mentioned here to help remind the reader.

Lines 93-95: highlighting some of the values would be beneficial here rather than this blanket statement I think. It is quite interesting that the one of tropical forest biomes result for example is more than twice the global average!

Line 111-112: "contributes importantly to the ..." I think some kind of value/result would be beneficial to back up this statement. Similar with statement in following sentence, "effect is disproportionately large".

Line 138 – Are some of the differences likely to be down to the type of data being included in each dataset as well?

Fig 2 – "see figure 1 for more details". I think it would be useful to copy the relevant details into each figure legend, including the extended data figures.

Figure 4: the legend or icons are muddled here. Amphibians is stated as panel a but it has a reptile icon.

Discussion:

Line 252 – could it also be because protected areas cover the ranges of some very rare species and so we would expect little land use change or use intensity in those areas? Just a thought.

Methods:

Line 333 – remove "We".

Line 338 – here and in others, I think the titles are missing what was intended to be symbols within some of the brackets.

Line 392 – I found this section rather challenging to follow. The description of what was done for each of the two intensity sets is very mixed together and it is not easy to distinguish between the two. It might help to clarify the differences between set 1 and set2 if these were separated out more in the section titled “Continuous land use intensity indices”. This section is still quite hard to follow and the differences between the two sets not easy to keep in mind. Perhaps a table outlining the two would be beneficial.

Line 455 – has the grazing metric used for Set 1 been mentioned before this?

Line 544 – looks like there is a missing variable in this sentence.

Line 554 – again variable missing and duplication on “in”.

Reviewer #3 (Remarks to the Author):

I am very pleased to see how the authors have addressed my comments to the previous version of the manuscript. I have no further comments.

Reviewer #1 (Remarks to the Author):

The authors have made a number of changes to the manuscript that help to clarify the methods, and now also highlight the two intensity indicator separately and compare their results in the main text and discussion. This helps to show the more novel element of the work more prominently.

Just a thought, but would the authors be able to provide usable maps of their two use intensity indicators, perhaps as a data product, so that they can be used in the future by others? I know I would be able to find a use for them!

Based on the data availability policy of Nature Communications, we deposited all relevant data on the Zenodo repository. This includes the maps of intensity indicator 1, however, indicator set 2 is based on already published data and may, therefore, not be provided by us; the new data availability statement should make it fairly easy, though, to access the relevant data.

We hope you understand that we would like to keep the HANPP-based intensity indicator set 1 under restricted access. The reason for this is that we are considering to publish these data in a separate publication in the future. However, we will grant access to these data upon request to the corresponding author.

Please refer to the relevant sections in the manuscript and data availability statement for further information.

Abstract:

Line 29: Line starting “Land-use intensity contributes...”. It feels like this sentence is stating 2 unrelated elements. Consider editing. I think removing the “ although large areas of the globe are still used with low intensity” will help.

We rephrased this sentence to: “Given the large fraction of global land currently used under low land-use intensity, we found its contribution to biodiversity loss to be substantial (~25%)”.

Mentioning the two sets of intensity indicators being tested and the data they focus on would help to present the novelty of this piece in the abstract. At the moment, very little of the abstract focuses on the intensity part.

Thanks for mentioning this and for the allowance of more words for the abstract. We rewrote parts of the abstract and expanded it to better include the two intensity indicator sets.

Introduction:

The introduction, specifically paragraphs 1 and 2, does very little to set the scene before diving straight into the description of the main methods. I think it could be beneficial to better outline why including intensity is so important, why only coarse measures have been used so far, and the fact that intensity can relate to so many variables. Some of these points are interspersed into the third paragraph so could be incorporated earlier.

Thanks for pointing this out. We moved some of this information from the third paragraph into the first paragraph. Specifically, the last part of the first paragraph now reads: “However, an

understanding of the effects of LU-intensity on biodiversity is critical, as LU intensification is expected to become pivotal in the future due to the increasing demands for land products and the simultaneous mandate to safeguard remaining pristine ecosystems^{10,11}. Due to the multidimensional nature of LU-intensity⁹, as well as large data uncertainties related to it^{12,13}, its impacts on global biodiversity could so far not be quantified satisfactorily. Here, we fill this gap and disentangle the contribution of LU-intensity from total biodiversity losses caused by LU practices worldwide with the help of new methodology”.

There is very little in terms of hypotheses or questions that you are trying to answer. I think that framing in this way would help with the structure. You state that you will “fill these gaps” but there is little linkage between the gaps, the methods and the results.

In order to link the research gaps and results better, and to introduce more general research questions, we added the following statement to the end of the second paragraph: “Here, we expand the cSAR approach to fill these gaps by increasing spatial resolution and explicitly including LU-intensity. Based on this approach, we answer how high the contribution of LU-intensity to total biodiversity loss is, and which individual species face regional extinction due to LU in general”.

Line 43: Another global study that includes use intensity is the Newbold et al paper that you use in the methods. This could be added as another reference here along with ref 10. It is still a coarse measure of intensity.

You are right, we included this reference here.

Line 67: remove misplaced “is”

Done.

Line 67: around here, it could be useful to add something stating if/when averages across these two sets are considered.

There is no statement regarding the comparison of the two indicators or what differences might be expected. This would help set up the discussion of regional differences that appear later.

We included the following sentence to clarify how the two sets are incorporated: “We use average effects of LU-intensity across both sets to estimate its general influence on biodiversity, and additionally explore global and regional differences between the two sets”. We had no a priori expectation about the differences between the two sets and therefore refrain from stating that somewhere in the manuscript.

Results:

Line 90: I think the land use types can be mentioned here to help remind the reader.

Done.

Lines 93-95: highlighting some of the values would be beneficial here rather than this blanket statement I think. It is quite interesting that the one of tropical forest biomes result for example is more than twice the global average!

Good idea. We inserted the sentence: “Especially tropical dry broad-leafed forests, temperate broad-leafed forests and Mediterranean forests show high species losses, each having, on average, twice as high losses than the global average.”

Line 111-112: “contributes importantly to the ...” I think some kind of value/result would be beneficial to back up this statement. Similar with statement in following sentence, “effect is disproportionately large”.

We improved these two sentences by changing them to: “Management of unconverted ecosystems contributes importantly to the LU-intensity effect, with livestock grazing on natural grazing land and wood extraction from primary forests contributing 1.54-1.57 and 0.89-0.98%-points to the total LU-intensity effect of ~3.55%, respectively. Although the majority of these unconverted, but used ecosystems are managed with low intensity (Supplementary Figure 3), their contribution to the total intensity effect is still disproportionately large because of the large area they cover globally (Supplementary Figure 4)”.

Line 138 – Are some of the differences likely to be down to the type of data being included in each dataset as well?

We included the half-sentence: “[...] or simply lack of knowledge or availability of coherent data products”.

Fig 2 – “see figure 1 for more details”. I think it would be useful to copy the relevant details into each figure legend, including the extended data figures.

Done, based on Fig.1 we added: “Species losses are calculated as sums across all 45 land-use types considered here in 5x5 arcmin landscapes due to land-use intensity alone, i.e. without the effect of land conversion. Numbers are proportions lost of the pristine species richness of each cell, estimated from species range maps and area-of-habitat approach. No lut = no land-use type found within the respective landscape”.

Figure 4: the legend or icons are muddled here. Amphibians is stated as panel a but it has a reptile icon.

Thanks for pointing this out, we corrected it accordingly (replaced the icons in the correct order).

Discussion:

Line 252 – could it also be because protected areas cover the ranges of some very rare species and so we would expect little land use change or use intensity in those areas? Just a thought.

Good point, we added the half-sentence: “[...] or that the AOH of these species are situated in protected areas where little to no LU is expected”.

Methods:

Line 333 – remove “We” .#

Done.

Line 338 – here and in others, I think the titles are missing what was intended to be symbols within some of the brackets.

We filled in all missing symbols in the section headers.

Line 392 – I found this section rather challenging to follow. The description of what was done for each of the two intensity sets is very mixed together and it is not easy to distinguish between the two. It might help to clarify the differences between set 1 and set2 if these were separated out more in the section titled “Continuous land use intensity indices”. This section is still quite hard to follow and the differences between the two sets not easy to keep in mind. Perhaps a table outlining the two would be beneficial.

An overview of the two sets is already given in Supplementary Data 2, and at the beginning of the section in question, in l 428, we included the sentence: “For an overview of which data products and assumptions went into the individual sets, please refer to Supplementary Data 2”.

We further added sub-section headings (Set 1 and Set 2) and made sure to only reference to the respective set within the sub-sections to prevent further confusion.

Line 455 – has the grazing metric used for Set 1 been mentioned before this?

Yes, in original line 431.

Line 544 – looks like there is a missing variable in this sentence.

That’s true, we inserted SR at the appropriate position.

Line 554 – again variable missing and duplication on “in”.

That’s true, too. We added the missing variable $LUI_{n,b}$.

Reviewer #3 (Remarks to the Author):

I am very pleased to see how the authors have addressed my comments to the previous version of the manuscript. I have no further comments.